# RBM7 deficiency promotes breast cancer metastasis by coordinating MFGE8 splicing switch and NF-kB pathway

**Fang Huang[1†], Zhenwei Dai[1†], Jinmiao Yu[1†], Kainan Wang[2†], Chaoqun Chen[1], Dan Chen[3], Jinrui Zhang[4], Jinyao Zhao[4], Mei Li[4], Wenjing Zhang[4], Xiaojie Li[5*], Yangfan Qi[4,6*], Yang Wang[1*]**

[1]Sino-US Research Center for Cancer Translational Medicine of the Second Affiliated Hospital of Dalian Medical University & Institute of Cancer Stem Cell, Dalian Medical University, Dalian, China; [2]Department of Oncology & Sino-US Research Center for Cancer Translational Medicine, the Second Affiliated Hospital, Dalian Medical University, Dalian, China; [3]Department of Pathology, the First Affiliated Hospital of Dalian Medical University, Dalian Medical University, Dalian, China; [4]Institute of Cancer Stem Cell, Dalian Medical University, Dalian, China; [5]Department of Prosthodontics, College of Stomatology, Dalian Medical University, Dalian, China; [6]Soochow University Cancer Institute, Suzhou, China

**\*For correspondence:**
xiaojieli0504@dmu.edu.cn (XL);
yangfanqi@dmu.edu.cn (YQ);
yangwang@dmu.edu.cn (YW)

[†]These authors contributed
equally to this work

**Competing interest:** The authors
declare that no competing
interests exist.

**Reviewing Editor:** Yongliang
Yang, Dalian University of
Technology, China

**Abstract** Aberrant alternative splicing is well-known to be closely associated with tumorigenesis of various cancers. However, the intricate mechanisms underlying breast cancer metastasis driven by deregulated splicing events remain largely unexplored. Here, we unveiled that RBM7 is decreased in lymph node and distant organ metastases of breast cancer as compared to primary lesions and low expression of RBM7 is correlated with the reduced disease-free survival of breast cancer patients. Breast cancer cells with RBM7 depletion exhibited an increased potential for lung metastasis compared to scramble control cells. The absence of RBM7 stimulated breast cancer cell migration, invasion, and angiogenesis. Mechanistically, RBM7 controlled the splicing switch of MFGE8, favoring the production of the predominant isoform of MFGE8, MFGE8-L. This resulted in the attenuation of STAT1 phosphorylation and alterations in cell adhesion molecules. MFGE8-L exerted an inhibitory effect on the migratory and invasive capability of breast cancer cells, while the truncated isoform MFGE8-S, which lack the second F5/8 type C domain had the opposite effect. In addition, RBM7 negatively regulates the NF-κB cascade and an NF-κB inhibitor could obstruct the increase in HUVEC tube formation caused by RBM7 silencing. Clinically, we noticed a positive correlation between RBM7 expression and MFGE8 exon7 inclusion in breast cancer tissues, providing new mechanistic insights for molecular-targeted therapy in combating breast cancer.

## eLife assessment

This study presents a rather **valuable** finding on the RBM7 function in spicing regulation and uncharacterized role of MFGE8 splicing alteration in breast cancer metastasis. The evidence supporting the claims of the authors is **solid**. The work will be of broad interest to clinicians, medical researchers and scientists working on breast cancer.

## Introduction

Alternative splicing (AS) emerges as a substantial contributor to the production of a myriad of transcripts from limited genes, which enriched protein variation and phenotypic diversity in multicellular organisms. Pre-mRNA splicing reactions require a step-wise assembly of spliceosome, a megadalton complex that recognize consensus regulatory sequences, including 5' and 3' splice sites marking exon-intron boundary, and the branch point site (BPS), to accomplish exon definition (*Shi, 2017*; *Wilkinson et al., 2020*). While a splice site is suboptimal, AS emerges under control of inherent sequence acting in cis, namely exonic or intronic splicing enhancer or silence, and the cognate trans-acting splicing factors, which can strengthen or weaken the recognition of splice sites by spliceosome.

It is well documented that AS dysregulation leads to multiple human diseases, including cancer (*Montes et al., 2019*; *Singh and Cooper, 2012*). Tumorigenicity has been linked with cancer-specific splicing switch arising from either cis-elements mutations or deregulated splicing regulatory factors. In some cases, splicing factors, particularly spliceosomal components SF3B1 and U2AF1, are frequently mutated across diverse cancer types (*Baralle and Baralle, 2005*; *Harbour et al., 2013*; *Smith et al., 2019*; *Cheruiyot et al., 2021*). For instance, recurrent SF3B1 mutations led to retention of intronic region in BRD9 pre-mRNA, conferring loss of core subunit function to the ncBAF chromatin remodeling complex and metastatic advantage to melanoma cells (*Inoue et al., 2019*). Another layer of regulation involves the abnormal expression of splicing factors derived from altered regulatory pathway. As an archetypal example, a set of members in SR protein family such as SRSF1 and SRSF3 are upregulated in breast tumors, lung tumors, glioma and exhibit oncogenic ability via altering AS products (for example, Bcl-x, MYO1B, NUMB, and p53; *Anczuków et al., 2015*; *Zhou et al., 2019*; *Tang et al., 2013*; *Zhou et al., 2020*). Otherwise, many splicing factors such as RBM (RNA binding motif) proteins can either functions as oncoproteins or tumor suppressors in cancer-type-dependent manner (*Li et al., 2021*), which may be attributed to tissue-specific AS patterns.

Breast cancer is the most common diagnosed malignancy and remain the second leading cause of cancer mortality (over 41,000 deaths annually in the USA) among females worldwide (*Siegel et al., 2019*). Metastasis to vital organs accounts for approximately 90% cancer-related death. The 5-year overall survival rate of patients with metastatic breast cancer is drastically decreased to about 25%, as compared to the primary cancer patients' prognosis with a survival rate greater than 80% (*Rabbani and Mazar, 2007*; *Valastyan and Weinberg, 2011*). Despite advances in early diagnosis and comprehensive interventions for breast cancer treatment, limited effective strategy against metastasis has been an outstanding challenge for improving prognostication of patients. This highlights the urgent need for deepened understanding of mechanisms underlying development of metastasis. Tumor metastasis is multi-step process involving a sequential cascade of events, such as epithelial-mesenchymal transition (EMT), local invasion from primary tumor site, lymphogenic or hematogenic dissemination, cancer cells seeding and metastatic lesions formation (*Liang et al., 2020*). AS dysregulation has been implicated in multistage process of metastasis. For example, the switch of CD44 or FGFR2 splicing isoforms can lead to EMT phenotype (*Chen et al., 2018*; *Warzecha et al., 2009*). VEGF-A or XBP1 splicing transition is tightly linked to angiogenesis in cancer (*Harper and Bates, 2008*; *Zeng et al., 2013*). On account of the phenotypic plasticity in metastatic cells conferred by alternative splicing, further investigation in potential role of critical splicing regulatory factors and the associated splicing events in breast cancer metastasis is warranted.

RNA-binding proteins (RBPs) are a large class of conserved proteins that participate in cancer development through precisely control RNA metabolism, including RNA splicing. RBM7, a member of RBP family, is commonly known to exert RNA surveillance activity as a component of the human nuclear exosome target (NEXT) complex (*Meola et al., 2016*). Aberrant expression of RBM7 is not only associated with lung fibrosis onset and motor neuronopathy, but also impacts cancer cell proliferation (*Fukushima et al., 2020*; *Xi et al., 2020*). Previous findings have established a direct connection of RBM7 with spliceosome factors SF3B2 and SF3B4 (*Falk et al., 2016*), yet whether RBM7 is involved in the regulation of AS remain elusive. In this study, we report that RBM7 is frequently reduced in a subset of breast cancer metastases from lymph nodes lesions and distant organ as compared to primary tumors and its low expression predicts dismal survival outcomes of patients. Breast cancer cells with loss of RBM7 gain enhanced aggressive capability and metastatic potential. Mechanistically, RBM7 ablation enhanced the splicing switch of MFGE8 pre-mRNA from canonical isoform MFGE8-L to a truncated isoform MFGE8-S. Ectopic expression of MFGE8-L caused marked suppression of

migratory and invasive ability of breast cancer cells, whereas MFGE8-S showed opposite function/disabled such function. At another layer, RBM7 discordantly regulated the phosphorylation of p65. In concert, NF-κB inhibitor abolished pro-angiogenesis effect of RBM7 ablation in breast cancer. Above all, our findings uncovered a previously unidentified role of RBM7 in breast cancer metastasis via coupling MFGE8 splicing switch and NF-κB signaling activation, suggesting a new therapeutic target and prognostic predictor for breast cancer.

## Results

### RBM7 is reduced in metastatic lesions of breast cancer and positively correlated with patients' prognosis

To determine the clinical relevance of RBM7 in breast cancer, we assessed the correlation between RBM7 and important clinical outcomes. Analysis of TCGA, GEO and the METABRIC breast cancer datasets showed that RBM7 expression positively correlates with overall survival of breast cancer patients (*Figure 1A* and *Figure 1—figure supplement 1A-B*). RBM7 low expression predicts the reduced disease-free survival in patients with breast cancer (*Figure 1B* and *Figure 1—figure supplement 1C*). Congruently, we found RBM7 is decreased in breast cancer compared to normal tissues (*Figure 1C* and *Figure 1—figure supplement 1D*). Further dissection in a cohort of breast cancer patients from the cBioPortal unveiled cases with lymph node metastasis have lower expression of RBM7 as compared to cases without metastasis (*Figure 1D*). We next performed immunohistochemistry (IHC) detection of RBM7 in breast cancer tissue microarrays. The results showed that patients bearing clinical stage III tumors have significantly decreased RBM7 expression than patients bearing I and II clinical stage tumors (*Figure 1E*). In tissue chip containing triple negative breast cancer, the more aggressive and mesenchymal breast cancer subtype, RBM7 exhibits reduced protein levels in tumors as compared to paracancer tissues (*Figure 1F*). In addition, we collect specimen consisted of tumor tissues from 21 breast primary cancer, in 9 cases, from metastatic loci and took advantage of IHC analysis to determine the RBM7 expression. Most (25% strong positive;67% moderate positive) of primary tumor samples displayed a positive IHC staining for RBM7, whereas most of lung/liver metastatic lesions showed weak or negative RBM7 expression (*Figure 1G*). We also obtained paired primary breast cancer and lymph node metastatic loci and found, when compared to the corresponding primary tumor, the expression of RBM7 is significantly decreased in lymph node metastases (*Figure 1H–I*). These results associated RBM7 with breast cancer metastasis, prompting us to further investigate the functional role of RBM7 in metastatic processes of breast cancer.

### RBM7 silencing provokes metastatic potential of breast cancer

In light with the above findings, we generated RBM7-silenced breast cancer cells using lentiviral expression of short hairpin RNA (shRNA) followed by profiling high-throughput transcriptome by RNA sequencing. Strikingly, Gene Ontology and pathway analysis displayed a wide range of differentially expressed genes were enriched in extracellular matrix organization, cell migration and angiogenesis (*Figure 2A–B*), which have been characterized as crucial processes in tumor metastasis. We then conducted qRT-PCR analysis of theses metastasis-associated genes in RBM7-depleted breast cancer cells and verified that they were up-regulated upon RBM7 knockdown shown as heatmap representation (*Figure 2C*). The lung metastasis is one of the most prevalent distant metastases that breast cancer is prone to and associated with poor prognosis of patients (*Medeiros and Allan, 2019*). Thus, we performed tail-vein transfer model to evaluate the potential impact of RBM7 on pulmonary metastasis of breast cancer cells. $6 \times 10^5$ 4T1 cells, the mouse breast cancer cell line simulating stage IV human breast tumor, with stable RBM7 silence or control vector were implanted into BALB/c mice via tail vein. RBM7-silenced 4T1 cells were able to develop more spontaneous lung metastases than control cells as judged by pulmonary nodules number and haematoxylin-eosin (HE) staining of lung tissues (*Figure 2D–F* and *Figure 2—figure supplement 1A*). Cancer metastasis is comprised of a complex cascade of events, such as migration, invasion and angiogenesis, thus we next to determine which process is mediated by RBM7-loss in facilitating breast cancer metastasis. We utilized transwell chamber with or without matrix coat and would healing assays to examine the potential function of RBM7 in migration and invasion capabilities and found the depletion of RBM7 led to a visible increase in mobility and invasiveness of both the triple-negative breast cancer (TNBC) cell line MDA-MB-231,

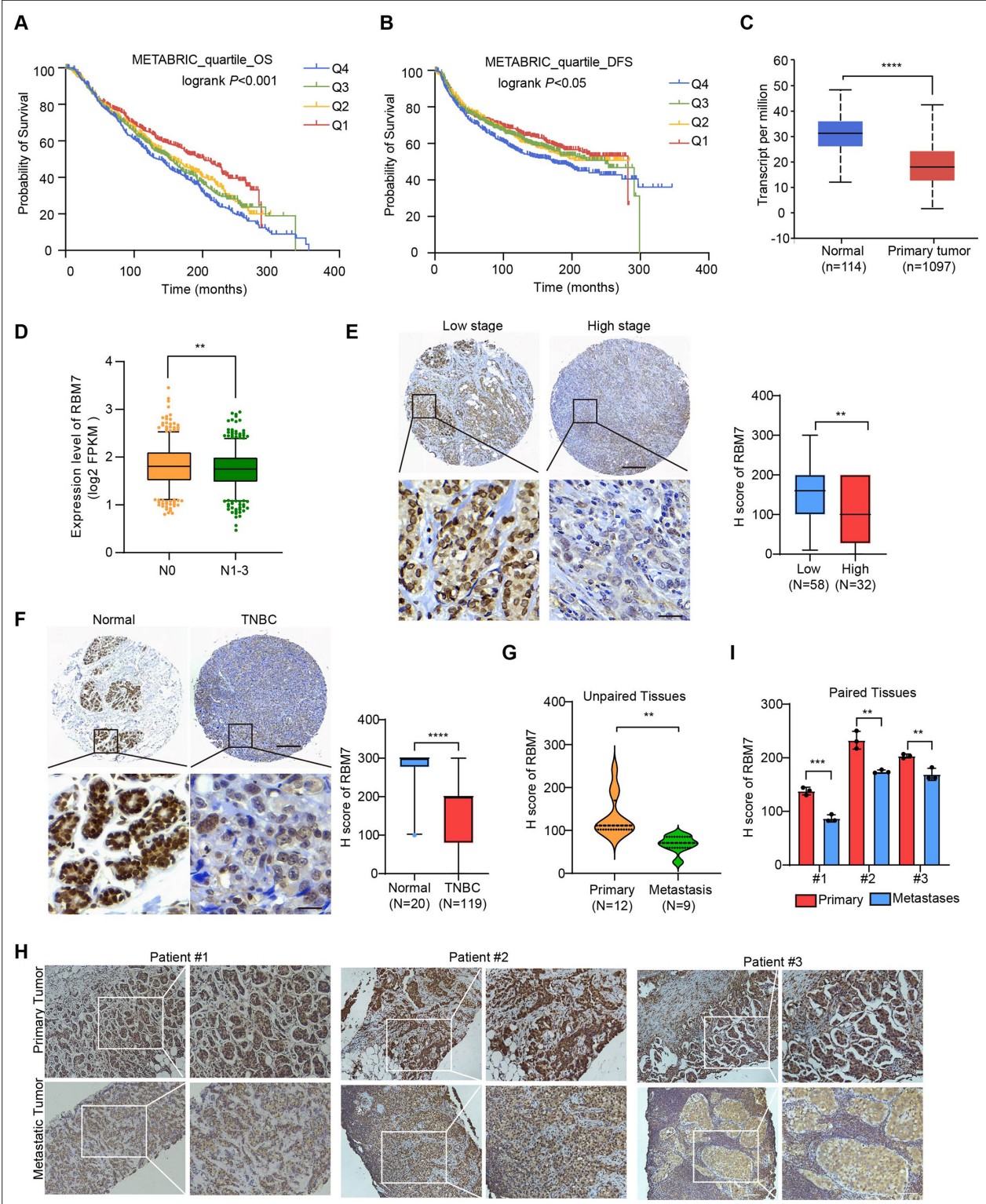

**Figure 1.** Decreased RBM7 expression was associated with poor prognosis of breast cancer. (**A–B**) The correlation between RBM7 mRNA expression and overall survival (OS) or disease-free-survival (DFS) of breast cancer patients (n=1980) was analyzed based on the METABRIC dataset. The samples were divided into four equal parts, containing lower quartile Q1, median quartile Q2, upper quartile Q3 and higher quartile Q4 according to the expression of RBM7. Data are presented as means ± SD and p values were obtained by Mantel-Cox log-rank test. (**C**) The expression of RBM7 in primary breast carcinoma (n=1097) compared to normal tissues (n=114) was analyzed through UALCAN dataset. (**D**) Analysis of RBM7 expression in breast cancer lymph node metastases in comparison with breast cancer tissues with no lymph node metastasis based on TCGA dataset. BRCA samples were

*Figure 1 continued on next page*

*Figure 1 continued*

classified into N0 (No regional lymph node involvement) (n=515), and metastases in lymph node (**N1–N3**) (n=565). (**E**) Representative IHC images of RBM7 staining for patients at high (n=32) or low (n=58) clinical stages on tissue microarray of breast cancer specimens. Scale bars = 300 μm (top) or 30 μm (bottom). (**F**) Representative images of the IHC staining of RBM7 in a tissue microarray containing triple-negative breast cancer (n=119) and para-carcinoma tissues (n=20). Scale bars = 300 μm (top) and 30 μm (bottom). (**G**) Quantitative analysis of RBM7 expression according to IHC staining scores in primary breast cancer (n=12) and distant metastases (breast cancer lung metastases, n=4; breast cancer liver metastases, n=5). (**H–I**) Representative IHC images and quantitative analyses for RBM7 staining in 3 paired primary breast cancer and lymphatic metastases are shown. Scale bar = 100 μm. Data are presented as means ± SD and p values were obtained by unpaired Student t test (**C–I**). **p<0.01, ***p<0.001, ****p<0.0001.

The online version of this article includes the following figure supplement(s) for figure 1:

**Figure supplement 1.** The low expression of RBM7 in breast cancer was positively correlated with poor prognosis of breast cancer patients.

estrogen receptor positive breast cancer cell line MCF7 and 4T1 cells (*Figure 2G–H* and *Figure 2—figure supplement 1B-E*). Conversely, RBM7 overexpression notably repressed MDA-MB-231 and MCF7 cells to migrate or invade across transwell inserts (*Figure 2I–J*). Similar results were obtained in two additional TNBC cell lines BT549 and HCC1937 (*Figure 2—figure supplement 1F-G*), which proved that RBM7 possessed the capability to suppress the migration and invasion of breast cancer cells. To examine the impact of RBM7 on angiogenesis in breast cancer, we cultured human umbilical vein endothelia cells (HUVEC) with condition medium obtained from culture supernatants of RBM7-depleted or scramble shRNA transduced cells for tube formation analysis. Compared to the scramble control, HUVEC incubated with culture medium from RBM7-knockdown breast cancer cells exhibited accelerated tube formation length and branch junction (*Figure 2K*), suggesting the inhibitory effect of RBM7 on breast cancer angiogenesis. Taken together, the above data demonstrated RBM7 as tumor suppressor by regulating breast cancer migration, invasion and angiogenesis.

## RBM7 exhibits splicing regulatory function in breast cancer

In previous studies, RBM7 is commonly characterized as an RNA-binding constituent of trimeric NEXT complex, which directs the RNA exosome to non-coding transcriptome for degradation (*Lubas et al., 2015*). However, the mechanistical role of RBM7 in cancer remain largely unknown. Intriguingly, we observed that, attributed to the ablation of RBM7, almost 551 splicing events occurred significant decrease in PSI (percent spliced in), whereas 289 splicing events endured increased PSI (*Figure 3A*). The changed events were divided among various types of AS (alternative splicing), including SE (skip-ping exon), IR (intron retention), A5E (alternative 5' splice site exon), A3E (alternative 3' splice site exon), and MXE (mutually exclusive exons), of note, SE events were the most prominently regulated (*Figure 3B*). A majority of AS events in particular of SE and A5E category were in decrease of PSI upon RBM7 depletion (*Figure 3C–D*), indicating that the presence of RBM7 may promote the usage of alternative splice sites of SEs or A5Es. Subsequently, we extracted the candidate binding motifs which reside within or near cassette exons or A5E and found UUUCUU motif, simiar to RBM7 binding site (polyU) were enriched in RBM7 positively regulated AS events to a greater extent that control exons unimpacted by RBM7 (*Figure 3E*), raising the possibility that RBM7 may serve as a splicing activator through directly tethering cis-element in pre-mRNA. This is coherent with the nuclear localization of RBM7 in breast cancer cells (*Figure 3—figure supplement 1A*). In order to explore the splicing regulation activity of RBM7 in distinct pre-mRNA context, we validated a series of RBM7-significantly changed splicing events across different AS types according to the RNA-seq results. The RT-PCR analysis demonstrated that the PSI of selected gene related to cancer progression was either increased or decreased endogenously upon RBM7 depletion (*Figure 3F* and *Figure 3—figure supplement 1B*). Considering that the AS is a sophisticated multi-step reaction that is regulated by both pre-mRNA-interacting splicing factor and other assistant factors in a context-dependent manner (*Wright et al., 2022*), we doubted the divergent effect of RBM7 on distinct splicing targets was due to a more complicated mechanism. Gene ontology (GO) enrichment analysis identified a majority of genes undergoing AS alterations were involved in cytoskeleton assembly and organization (*Figure 3G*). While globally analyzing the network enrichment of RBM7-regulated downstream splicing events, we observed its targets are functionally connected to well-linked network containing gene subgroups (*Figure 3H*), which are associated with biological processes in tumor cell metastasis, again suggesting the cancer-related function of RBM7.

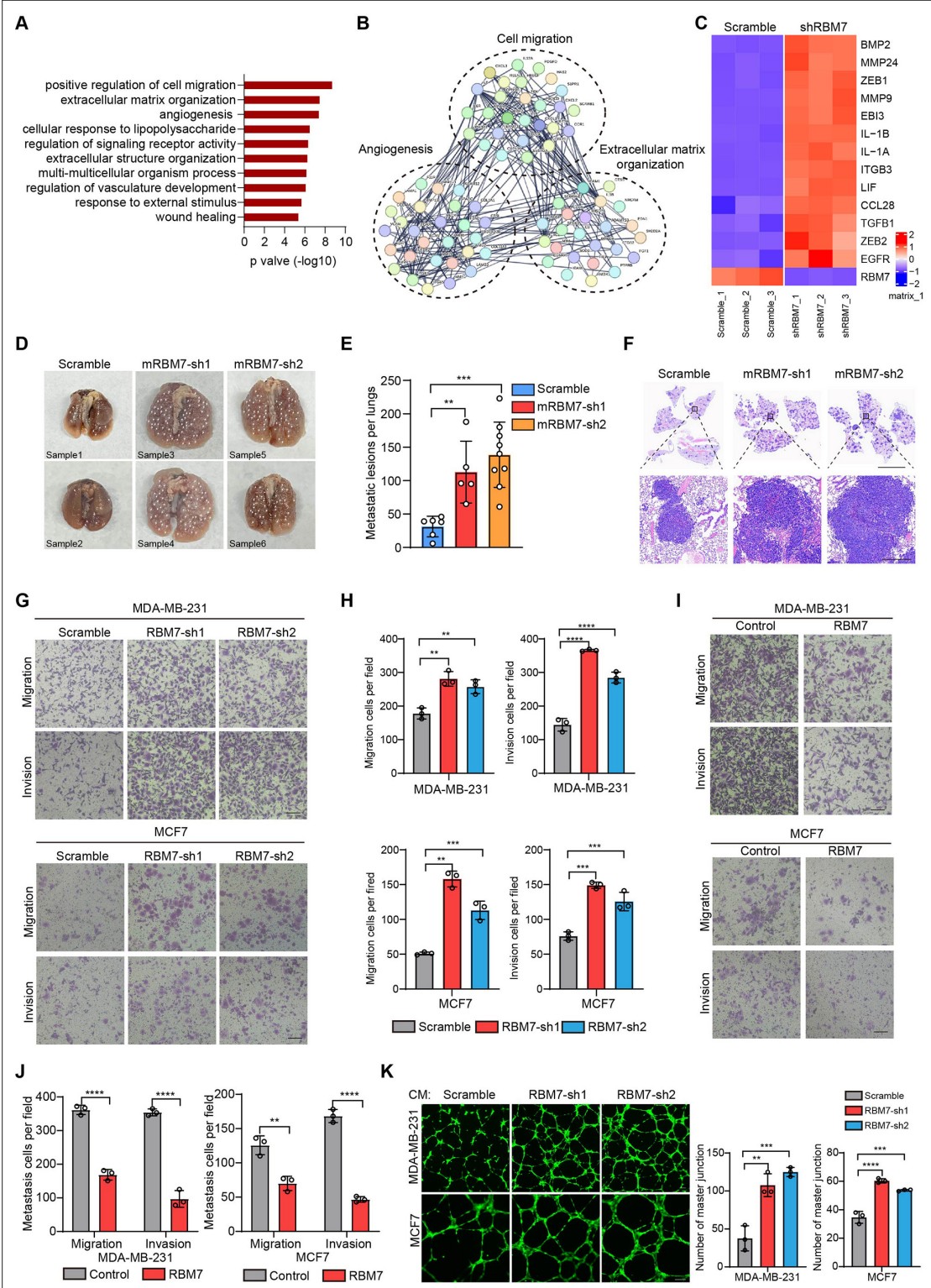

**Figure 2.** RBM7 negatively regulated breast cancer metastatic potential. (**A**) Gene Ontology analysis showed the significantly affected biological process upon RBM7 knockdown in MDA-MB-231 cells. (**B**) Functional association network of RBM7-regulated targets. The genes were analyzed using the STRING database, and subgroups are marked according to their function. (**C**) A heatmap showing the qRT-PCR analysis of differentially expressed genes upon RBM7 knockdown in breast cancer cells from three biological replicates. (**D–F**) 4T1 cells without or with RBM7 knockdown were injected into tail vein of immunodeficient BALB/c mice to establish a lung metastasis model. Spontaneous pulmonary metastases were assessed after about 3 weeks, the white arrow indicates lung lesions macroscopically (**D**). (**H and E**) stained lung sections were quantified for the number of spontaneous

*Figure 2 continued on next page*

*Figure 2 continued*

metastatic lesions from BALB/c mice (n=6, 5 and 9). Data are presented as mean ± SD and p values were determined by one-way ANOVA with Dunnett's multiple comparisons test (**E**). Representative images of H&E stained lung sections are shown. Top panel scale bars indicate 7 mm; bottom panel scale bars indicate 300 µm (**F**). (**G–J**) The metastatic ability of breast cancer cells with RBM7 depletion (**G**) or ectopic expression of RBM7 (**I**) were evaluated by transwell assay. Scale bars: 100 µm. Data are mean ± SD from three random fields. p values were determined by one-way ANOVA with Dunnett's multiple comparisons test (**H**) or unpaired Student's T test (**J**). (**K**) Representative images of the tube formation of HUVEC cells treated with conditional medium from RBM7-KD cells or control cells for 12 hr. Scale bars: 100 µm. Quantification of number of junctions formed by HUVEC was calculated by Image J software. **p < 0.01, ***p < 0.001, ****p < 0.0001.

The online version of this article includes the following source data and figure supplement(s) for figure 2:

**Figure supplement 1.** RBM7 depletion promoted breast cancer cell migration and invasion.

**Figure supplement 1—source data 1.** Original file for the western blot analysis in *Figure 2—figure supplement 1D* (anti-RBM7 and anti-tubulin).

**Figure supplement 1—source data 2.** PDF containing *Figure 2—figure supplement 1D* and original scans of the relevant western blot analysis (anti-RBM7 and anti-tubulin) with highlighted bands and sample labels.

**Figure supplement 1—source data 3.** Original file for the Western blot analysis in *Figure 2—figure supplement 1F* (anti-RBM7 and anti-tubulin).

**Figure supplement 1—source data 4.** PDF containing *Figure 2—figure supplement 1F* and original scans of the relevant western blot analysis (anti-RBM7 and anti-tubulin) with highlighted bands and sample labels.

## RBM7 depletion switches splicing outcomes of MFGE8 by decreasing MFGE8-L isoform

To ascertain the functional targets that explain the inhibitory role of RBM7 in breast cancer metastasis, we analyzed RNA immunoprecipitation sequencing data for RBM7 and whole-transcriptome sequencing data for breast cancer cells expressing two shRNAs targeting RBM7. Venn diagram of 8 hub genes obtained by overlapping three gene sets pointed forward a hotspot gene, namely MFGE8 (*Figure 4A*). RNA-seq data showed that the depletion of RBM7 resulted in a significant alteration in ratio of MFGE8 two isoforms, of which the canonical isoform (MFGE8-L) has been demonstrated to be correlated with tumor formation and aggressiveness of breast cancer (*Yang et al., 2011*). Importantly, the exon 7 located in F5/8 type C 2 domain, which is obligated to bind to phosphatidylserine, was lacked in the short isoform of MFGE8 (MFGE8-S) (*Figure 4B*). Further RT-PCR identification verified RBM7 ablation indeed potentiated the production of MFGE8-S in various TNBC cell lines (*Figure 4C*). Similar results were obtained from MCF7 cells, suggesting that RBM7 regulated the AS of MFGE8 independently of the ER status. In separate experiments, we transfected breast cancer cells with an exogenous MFGE8 splicing reporter containing exon 6–8 and 600 bp of flanking intron sequences to determine the splicing regulatory function of RBM7. Accordingly, the depletion of RBM7 still induced the splicing shift of the minigene reporter by elevating MFGE8-S variant (*Figure 4D*). To test whether such splicing regulatory effect was directly or indirectly, we analyzed the sequence of alternative exon 7 and the motifs nearby its 5' or 3' splice sites, and found two poly(U) motifs are positioned in proximal to the pseudo 3' splice site. Subsequent RT-PCR assay for the precipitation in RIP assays confirmed RBM7 could bind to the upstream sequence containing UUUCUU motifs adjacent to 3' splice site of MFGE8 cassette exon, but not another region nearby it (*Figure 4E*). To pinpoint the location for the potential cis-element for AS regulation by RBM7, we designed antisense oligonucleotides (ASOs), which have been chemically modified to avoid endogenous nuclease degradation without recruiting RNase H, to block the UUUCUU residues in proximity to intron6/exon7 junction. As shown in *Figure 4F*, when compared to control ASO, treatment with ASOs targeting the potential binding site (UUUCUU) of RBM7, contributed to the exclusion of exon7 of MFGE8. In separate experiments, the depletion of RBM7 induced the splicing shift of the minigene reporter by elevating MFGE8-S variant. While the binding motif was depleted or mutated, RBM7 failed to affect the splicing outcomes of MFGE8 (*Figure 3—figure supplement 1C*). Due to its close proximity to 3' splice site, UUUCUU residues bound by RBM7 might participated in spliceosome assembly on the upstream 3' splice site of exon7, which may explain why disruption of the motif led to almost complete exon7 skipping. Oue data suggest that RBM7 regulates the exon skipping of MFGE8 by directly binding to UUUCUU located at six nucleotides upstream of the 3' splice-site of exon7.

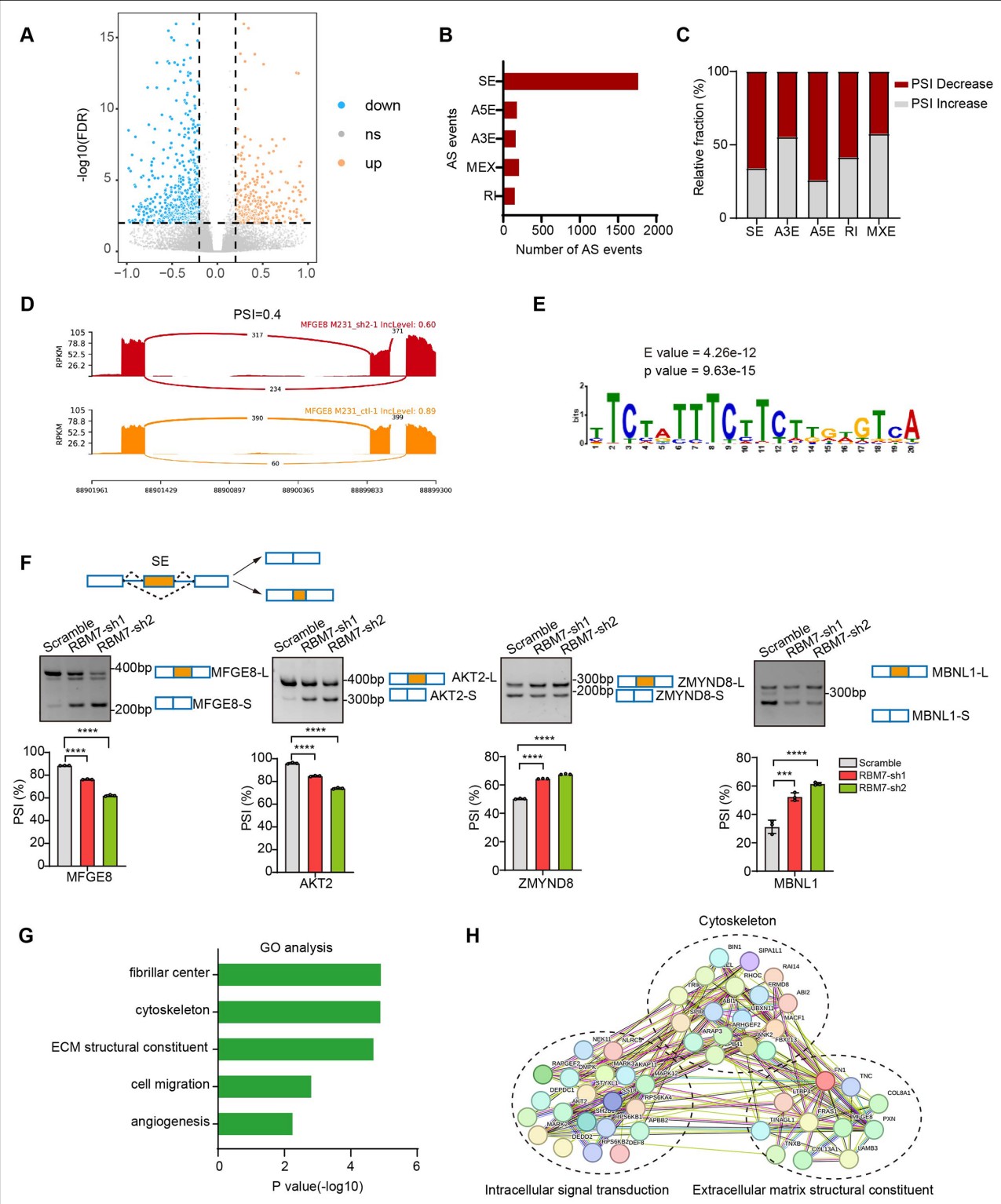

**Figure 3.** Global identification of alternative splicing events regulated by RBM7. (**A–E**) RNA-seq analysis was performed on RBM7-knockdown MDA-MB-231 cells or control cells, and the changes in splicing events were analyzed. (**A**) Volcano plot illustrating up-regulated/down-regulated alternative splicing events upon RBM7 depletion. (**B**) Comparison of the differential splicing events in five types of AS events affected by RBM7 depletion. (**C**) The relative fraction of splicing events undergoing percent spliced in (PSI) increase or decrease induced by RBM7 depletion. (**D**) Alternative splicing of MFGE8 was chosen to represent a decrease of PSI, and numbers of exon junction reads were indicated. (**E**) The hit top enriched motifs identified in the differential spliced genes regulated by RBM7. (**F**) RBM7-regulated exon skipping events were identified by semiquantitative RT-PCR using RBM7-

*Figure 3 continued on next page*

*Figure 3 continued*

depleted or control MDA-MB-231 cells (see also *Figure 3—source data 1*, *Figure 3—source data 2*). The mean ± SD of PSI from three independent experiments was plotted, with p values determined by one-way ANOVA with Dunnett's multiple comparisons test. (**G**) Gene Ontology analysis showed the significantly enriched biological processes of changed splicing events affected upon RBM7 knockdown in MDA-MB-231 cells. (**H**) KEGG pathway enrichment analysis of functional association network of the RBM7-controlled AS targets.

The online version of this article includes the following source data and figure supplement(s) for figure 3:

**Source data 1.** Original file for the RT-PCR analysis of splicing shifts in *Figure 3F* (MFGE8, AKT2, ZMYN8, or MBNL1).

**Source data 2.** PDF containing *Figure 3F* and original scans of the relevant RT-PCR analysis of splicing shifts (MFGE8, AKT2, ZMYN8, or MBNL1) with highlighted bands and sample labels.

**Figure supplement 1.** RBM7 had a potential role in regulating alternative splicing of breast cancer cells.

**Figure supplement 1—source data 1.** Original file for the RT-PCR analysis of MAP7D1 or HNRNPC splicing switch in *Figure 3—figure supplement 1B*.

**Figure supplement 1—source data 2.** PDF containing *Figure 3—figure supplement 1B* and original scans of the relevant RT-PCR analysis for splicing switch of MAP7D1 or HNRNPC variants with highlighted bands and sample labels.

**Figure supplement 1—source data 3.** Original file for RT-PCR analysis of splicing switch of MFGE8 minigene reporter in *Figure 3—figure supplement 1C*.

**Figure supplement 1—source data 4.** PDF containing *Figure 3—figure supplement 1C* and original scans of the relevant RT-PCR analysis of MFGE8 splicing switch with highlighted bands and sample labels.

## The short variant of MFGE8 abolishes the inhibitory function of its classical isoform on breast cancer cell migration and invasion

MFGE8 encodes a secreted protein, Milk fat globule-EGF factor 8 (MFG-E8), which participates in a variety of cellular processes, including mammary gland branching morphogenesis, angiogenesis, and mucosal healing, etc (*Raymond et al., 2009*). We applied MFGE8-L stably overexpressing- or depleted-breast cancer cells to transwell assays and observed the cellular migratory and invasive capabilities was visibly suppressed by MFGE8-L overexpression (*Figure 5—figure supplement 1A-B*). On the contrary, the knockdown of MFGE8-L provoked cell migration and invasion (*Figure 5—figure supplement 1C*). Strikingly, while MDA-MB-231 or HCC1937 expressing MFGE8-L showed a remarkable decrease in migratory and invasive ability compared to control vector, MFGE8-S not only totally abolished the inhibitory function of MFGE8-L but elicited modest pro-migration/ invasion actions on breast cancer cells (*Figure 5A* and *Figure 5—figure supplement 1D-E*). We also performed Fluorescent Gelatin Degradation Assays for investigating invadopodia formation and found MFGE8-L up-regulation suppressed breast cancer cells invasion through a layer of extracellular matrix, whereas breast cancer cells with ectopic expression of MFGE8-S acquired enhanced ability to degrade matrix and invasion (*Figure 5B*). Immunofluorescence assays revealed the MFGE8-L forms foci in cytoplasm, particularly enriched near cell-cell contact sites, whereas MFGE8-S is more evenly distributed in the nucleus and cytoplasm (*Figure 5C*). As shown in diagram of domain configuration for two variants, in comparison with MFGE8-L isoform containing N-terminal EGF-like domain and C-terminal twice-repeated F5/8 type C domains, the protein encoded by MFGE8-S loses the second F5/8 type C domain, implying a shortage of the function corresponding to membrane-binding (*Figure 5—figure supplement 1F*). The different structural bases of two isoforms may also be linked to their distinct cellular function in tumor invasion (*Figure 5—figure supplement 1G*). Furthermore, we compared transcriptome profiles between MFGE8-L and MFGE8-S ectopically expressing breast cells and found a set of functional genes were down-regulated by MFGE8-L, whereas this negative regulation was abrogated in response to MFGE8-S (*Figure 5—figure supplement 1H*). The differentially expressed genes between two splice variants were predominantly enriched in cell adhesion molecules and JAK-STAT cascade as revealed by KEGG pathway and Gene ontology analysis (*Figure 5D* and *Figure 5E*). Subsequent experimental validation demonstrated that the activation of STAT1 was hindered by MFGE8-L, but not affected by MFGE8-S (*Figure 5F*). Considering that STAT1 signaling activation plays an important role in tumor cell metastasis via cell adhesion molecules induction (*Marimuthu et al., 2022*; *Chen et al., 2019*), it is highly plausible that MFGE8-L, but not MFGE8-S, impeded breast cancer cell motility and invasive capability in a STAT1-dependent manner.

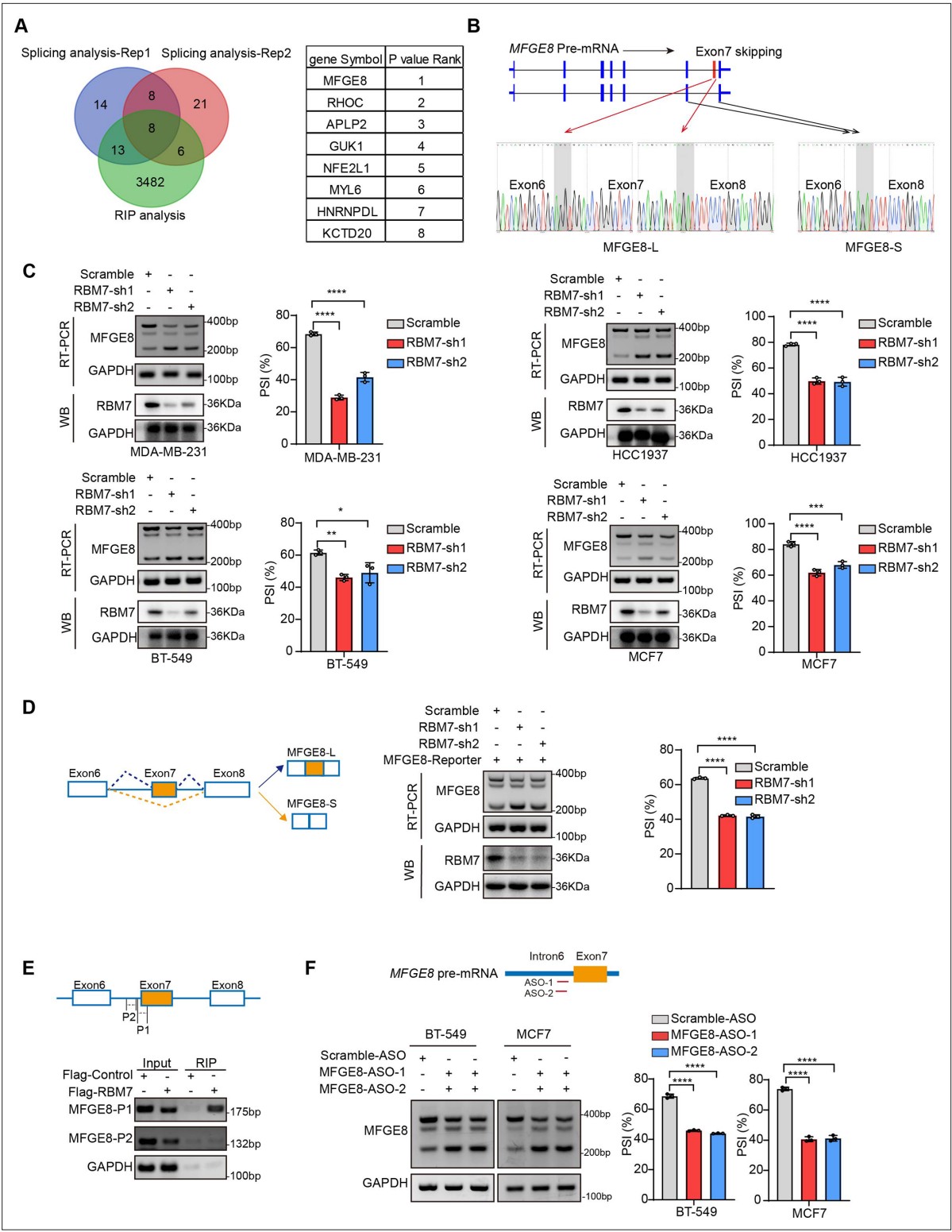

**Figure 4.** RBM7 knockdown promoted exon 7 skipping of MFGE8. (**A**) Comparison of RBM7-binding candidates from the RNA immunoprecipitation (RIP) sequencing (GSE144075 dataset) and two gene sets containing top 43 differentially AS genes in RNA-seq data of breast cancer cells expressing RBM7 shRNA1/2 as presented by venn diagram. The gene lists on the right were shown according to the FDR *P* value of top genes with significantly changed AS in RNA-seq. (**B**) Upper: schematics of human MFGE8 pre-mRNA (NM_005928.4). MFGE8-long isoform included the exon 7 (hereafter referred to as MFGE8-L), whereas it is skipped in MFGE8-short isoform (hereafter referred to as MFGE8-S). The arrow indicates the direction of transcription. Lower: primers were designed on exon 6/8, and RT-PCR was performed to identify PSI changes using RBM7-knockdown or control MDA-

*Figure 4 continued on next page*

*Figure 4 continued*

MB-231 cells. The base peak diagram of sanger sequencing of RT-PCR results showed the splicing junction sites (filled with gray). (**C**) Various breast cancer cells lines with stable depletion of RBM7 or control were constructed. Alternative splicing events of MFGE8 regulated by RBM7 were examined by semi-quantitative RT-PCR. The knockdown efficiency of RBM7 expression was detected by western blotting. (see also *Figure 4—source data 1*, *Figure 4—source data 2*). The mean ± SD of PSIs from three independent repeated experiments were plotted with p value calculated by one-way ANOVA with Dunnett's multiple comparisons test. (**D**) Left, Schematics of MFGE8 mini-splicing reporter including exons 6–8 and two flanking introns in the 600 bp region upstream and downstream of 5'/3' splice sites; middle, RBM7 stable knockdown or control cells were transfected with splicing reporter and collected for RNA extraction after 48 hr. The splicing changes of exogenous MFGE8 were detected by RT-PCR with specific primers on the reporter gene. The level of RBM7 protein was detected in MCF7 cells by western blotting (see also *Figure 4—source data 3*, *Figure 4—source data 4*). p value was calculated by one-way ANOVA with Dunnett's multiple comparisons test from three repeated experiments. (**E**) As shown in schematic diagram, orange box indicates cassette exon 7 and primers were designed in the putative binding regions P1 and P2. Binding of MFGE8 pre-mRNA with RBM7 was examined by RIP assay in HEK293T cells expressing Flag-RBM7. (see also *Figure 4—source data 5*, *Figure 4—source data 6*) (**F**) Upper: the red line in diagram indicates ASOs targeting region which contain UUUCUU residues; down: MCF7 and BT-549 cells were transfected with ASOs targeting MFGE8 pre-mRNA for 48 hr and then applied for RT-PCR identification (see also *Figure 4—source data 7*, *Figure 4—source data 8*). *p<0.05, **p<0.01, ***p<0.001, and ****p<0.0001.

The online version of this article includes the following source data for figure 4:

**Source data 1.** Original file for the RT-PCR analysis of MFGE8 splicing switch and western blot identification of RBM7 knockdown in various breast cancer cell lines in *Figure 4C*.

**Source data 2.** PDF containing *Figure 4C* and original scans of the relevant RT-PCR analysis for MFGE8 splicing switch and western blot identification of RBM7 knockdown with highlighted bands and sample labels.

**Source data 3.** Original file for the RT-PCR analysis of splicing switch in MFGE8 mini-splicing reporter and western blot identification of RBM7 knockdown in *Figure 4D*.

**Source data 4.** PDF containing *Figure 4D* and original scans of the relevant RT-PCR analysis of splicing switch in MFGE8 mini-splicing reporter and western blot identification of RBM7 knockdown with highlighted bands and sample labels.

**Source data 5.** Original file for the RT-PCR analysis of RBM7 protein- MFGE8 pre-mRNA binding from RIP assay in *Figure 4E*.

**Source data 6.** PDF containing *Figure 4E* and original scans of the RT-PCR analysis of RBM7 protein- MFGE8 pre-mRNA binding from RIP assay with highlighted bands and sample labels.

**Source data 7.** Original file for the RT-PCR analysis of MFGE8 splicing switch upon ASO treatment in *Figure 4F*.

**Source data 8.** PDF containing *Figure 4F* and original scans of the relevant RT-PCR analysis for splicing switch of MFGE8 variants with highlighted bands and sample labels.

## RBM7 depletion-stimulated aggressiveness of breast cancer is dependent on MFGE8 splicing switch and NF-kB pathway activation

Next, we sought to assess if MFGE8 splicing switch is responsible for acquisition of aggressive feature in RBM7-silenced breast cancer cells. Flag-tagged MFGE8-L or an empty vector was co-expressed in MDA-MB-231 cells together with RBM7 shRNA (*Figure 6—figure supplement 1A*). Following seeding into transwell chamber, cells transfected with RBM7 shRNA exerted a higher migratory and invasive capability than scramble control, consistent with above data. Importantly, the restoration of MFGE8-L expression diminished the acquired aggressive properties of MDA-MB-231 cells resulted from RBM7 depletion (*Figure 6A*). Similar results were obtained using RBM7-depleted MCF7 cells with or without MFGE8-L complement (*Figure 6A*). To corroborate the role of MFGE8 splicing shift in RBM7- regulated breast cancer cell invasion, we transfected triple-negative breast cancer cells with ASOs directed against RBM7-binding motif and examined the potential impact on cell aggressiveness. The results showed an obvious increase in exon7-skipped variant as compared to the scramble negative control ASOs (*Figure 6—figure supplement 1B*). Meanwhile, the migrative and invasive ability of breast cancer cells treated with ASOs was significantly boosted (*Figure 6B*), further suggesting that RBM7-knockdown stimulated aggressiveness of breast cancer at least partially relies on the control of MFGE8 AS. However, a series of shRBM7-upregulated secreted factors that have been reported to drive tumor angiogenesis (*Gopinathan et al., 2015*; *Fousek et al., 2021*; *Feng et al., 2019*), were not impacted by MFGE8-L/S (*Figure 6C* and *Figure 6—figure supplement 1C*). Condition medium derived from breast cancer cells overexpressing MFGE8-L/S showed negligible effect on HUVEC tube formation *Figure 6—figure supplement 1D*, implying that there exists additional mechanism underlying breast cancer metastasis attributed to RBM7 reduction.

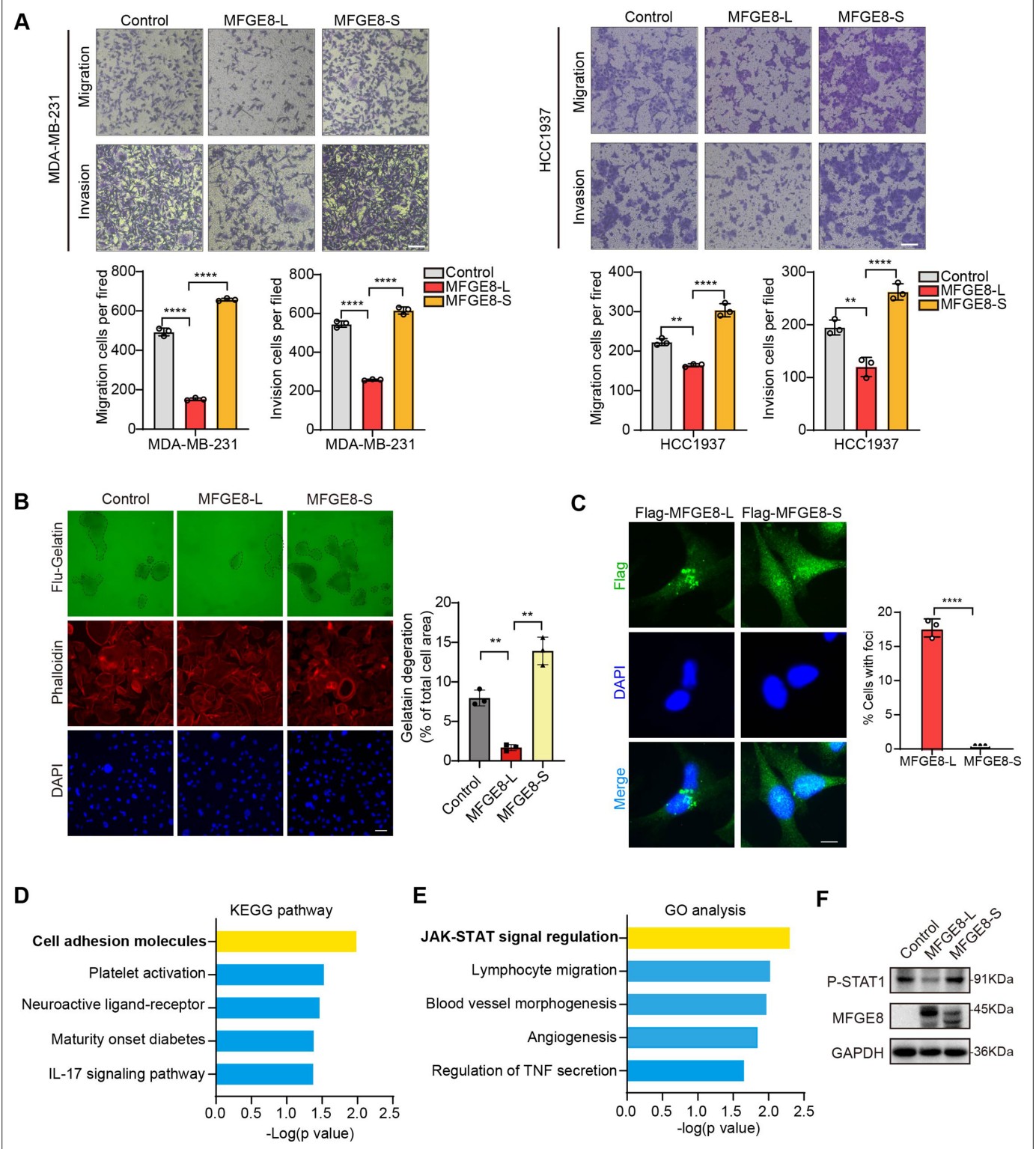

**Figure 5.** MFGE8-L inhibited breast cancer cell migration and invasion. (**A**) The function of MFGE8-L/S on breast cancer cell migration and invasion were evaluated by transwell assays. Scale bars: 100 μm. (**B**) Gelatin degradation assay was performed to test the effect of RBM7 knockdown on invadopodia function. A total of 10,000 cells were plated onto FITC-gelatin substrates (Green) and cultured for 48 hr. Representative images are shown (red, Cy3-phalloidin; blue, DAPI) and the degraded areas were quantified by Image J software. Scar bars = 50 μm. *P* values were determined by one-way ANOVA with Tukey's multiple comparisons test (n=3). (**C**) The subcellular localizations of MFGE8 isoforms. MDA-MB-231 cells were transfected with Flag-tagged MFGE8-L or MFGE8-S, and visualized with immunofluorescence assay. Scale bar = 10 μm. The percent of cells with foci were quantified and

*Figure 5 continued on next page*

*Figure 5 continued*

plotted (about 80 cells were captured and quantified in both samples). (**D–E**) The KEGG pathway and Gene ontology analysis of genes up-regulated by MFGE-S compared with MFGE-L. (**F**) The expression levels of p-STAT1(Tyr701) in HCC1937 cells overexpressing MFGE8-L/S was detected by western blotting. (see also *Figure 5—source data 1*, *Figure 5—source data 2*). p values were determined by one-way ANOVA with Tukey's multiple comparisons test. **p<0.01, ****p<0.0001.

The online version of this article includes the following source data and figure supplement(s) for figure 5:

**Source data 1.** Original file for the western blot analysis in *Figure 5F* (anti-P-STAT1, anti-MFGE8 and anti-GAPDH).

**Source data 2.** PDF containing *Figure 5F* and original scans of the relevant western blot analysis (anti-P-STAT1, anti-MFGE8 and anti-GAPDH) with highlighted bands and sample labels.

**Figure supplement 1.** MFGE8-L suppressed breast cancer cells migration and invasion.

**Figure supplement 1—source data 1.** Original file for the western blot analysis in *Figure 5—figure supplement 1A* (anti-MFGE8 and anti-GAPDH).

**Figure supplement 1—source data 2.** PDF containing *Figure 5—figure supplement 1A* and original scans of the relevant western blot analysis (anti-MFGE8 and anti-GAPDH) with highlighted bands and sample labels.

**Figure supplement 1—source data 3.** Original file for the western blot analysis in *Figure 5—figure supplement 1D* (anti-Flag and anti-GAPDH).

**Figure supplement 1—source data 4.** PDF containing *Figure 5—figure supplement 1D* and original scans of the relevant western blot analysis (anti-Flag and anti-GAPDH) with highlighted bands and sample labels.

Given that consecutive gene set enrichment analysis (GSEA) of highest-ranking gene sets based on transcriptomic profile of mRNA expression showed a signature for NF-κB pathway that was augmented upon depletion of RBM7 (*Figure 6D*), we suspected that the transcriptional activation of downstream genes in NF-κB signaling known to facilitate tumor angiogenesis might be involved in shRBM7-exerted pro-metastatic function. As expected, the phosphorylation of p65, a core transcription factor well-known to switch on NF-κB pathway, was notably enhanced by RBM7 knockdown. Meanwhile, IκBα phosphorylation that can blockade p65 activation was curbed upon RBM7 knockdown (*Figure 6E*). Conversely, RBM7 overexpression impeded the activation of p65 (*Figure 6—figure supplement 1E*). Yet MFGE8-L/S had no effect on phosphorylation of p65 or IκBα (*Figure 6—figure supplement 1F*). We found RBM7 depletion remarkably promoted the expression of IL-1β as judged by qPCR and ELISA assays (*Figure 6—figure supplement 1G-I*). IL-1β, commonly known as a pro-inflammatory cytokine, could bind to IL-1R and initiate a multistage enzymatic reaction that triggers the activation of NF-κB pathway (*Weber et al., 2010*; *Guo et al., 2024*). Thus, it is plausible the upregulation of IL-1β might be a causal factor in RBM7-depletion-induced activation of NF-kB signaling. We further carried out Matrigel tube formation assays and found NF-κB inhibitor PDTC drastically reversed the promotion of HUVEC angiogenesis by conditional medium harvested from RBM7-depleted cells (*Figure 6F–G*), implying that RBM7-p65 axis may participate in HUVEC angiogenesis through reprogramming the secretome of breast cancer cells. Similar effect was observed with respect to cell motility and invasion (*Figure 6—figure supplement 1J*). Taken together, these data above indicated that both MFGE8 splice switch and NF-κB pathway activation are required for mediating the pro-metastasis activity of RBM7 reduction in breast cancer.

## The PSI of MFGE8 exon 7 is decreased in breast cancer patients and positively correlated with RBM7 expression

To study the clinical relevance of MFGE8 isoform switch, we analyzed the PSI changes of cassette exon7 in TCGA breast cancer cohort and found exon7 skipping occurs more frequently in tumor than in normal tissues (*Figure 7A*). Consistently, in comparison with normal counterparts, the splicing of MFGE8 is shifted towards a reduced proportion of MFGE8-L across multiple cancer types, including LUAD, HNSC, COAD, etc (*Figure 7—figure supplement 1A*). RT-PCR assay was performed in surgical tumor specimens in breast cancer patients and showed increased proportion of exon7 exclusion in most tumor samples when compared to normal tissues (case#1: 86:94; case#2: 84:86; case#3: 79:85; case#4: 63:75; case#5: 69:93; case#6: 71:80; *Figure 7B*). On the other hand, the lymph node metastases contain a higher proportion of MFGE8 variant with skipped exon7 in comparison with paired primary tumor tissues (case#1: 80:95; case#2: 86:97; case#3: 84:90; case#4: 70:78; case#5: 83:89; *Figure 7C*). This is coherent with decreased RBM7 expression levels found in breast cancer with lymph node metastasis. In addition, patients with high expression of MFGE8 exon7 had a visibly better overall survival than patients with low expression (*Figure 7D*). We performed Pearson correlation test

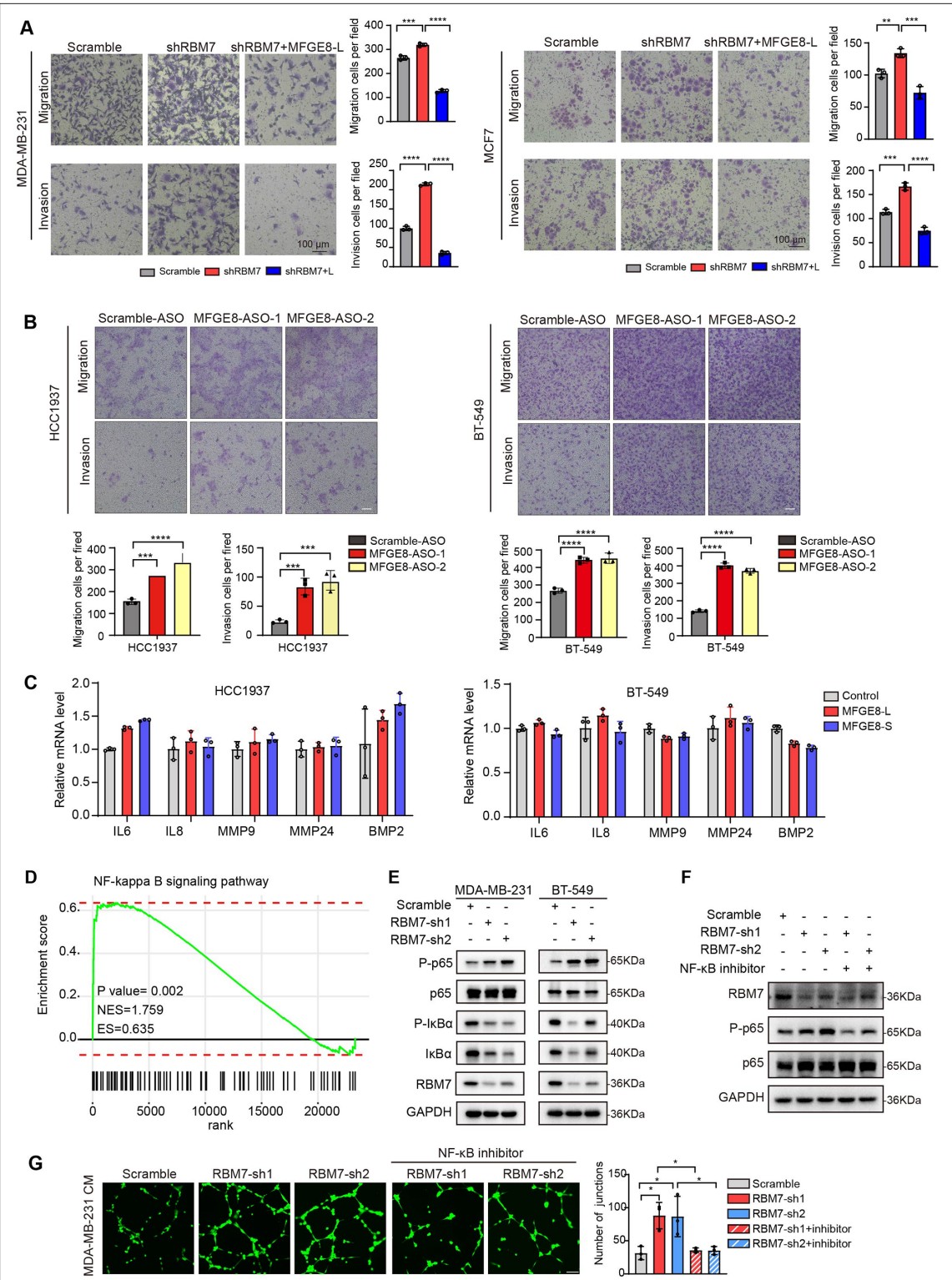

**Figure 6.** RBM7 knockdown enhanced aggressiveness of breast cancer relying on MFGE8 splicing switch to the short variant and NF-$\kappa$B pathway activation. (**A**) Breast cancer cells MDA-MB-231 and MCF7 were cotransfected with shRNA targeting RBM7 or control and shRNA-resistant Flag-MFGE8-L vectors and then subjected to transwell migration assay or invasion assay. Migrated or invaded cells were counted from random sites of the transwell with Image J. p values were determined by one-way ANOVA with Tukey's multiple comparison test. Scale bars: 100 μm. (**B**) Representative transwell analysis of migrative/invasive capability of breast cancer cells transfected with 500 nM ASO directed against RBM7-binding region in MFGE8 pre-mRNA. Cells migrating through transwell membrane with matrigel or not were imaged and representative images are shown (upper) along with

*Figure 6 continued on next page*

*Figure 6 continued*

quantification (down). Scale bars: 100 µm.p values were determined by one-way ANOVA with Tukey's multiple comparison test. (**C**) The expression of metastasis-related factors in HCC1937 and BT-549 cells stably expressing MFGE8-L, MFGE8-S or control vector was measured by real-time PCR assay. (**D**) NF-$\kappa$B signaling pathway was enrichment via KEGG analysis in RBM7 knockdown breast cancer cells. (**E**) Western blotting showed the expression of p65, p-p65, I$\kappa$B$\alpha$ and p-I$\kappa$B$\alpha$ in NF-$\kappa$B pathway in RBM7-depleted or control MDA-MB-231 and BT-549 breast cancer cells (see also *Figure 6—source data 1*, *Figure 6—source data 2*). (**F**) Western blotting was performed to test the expression of RBM7, p65 and p-p65 in RBM7-depleted MDA-MB-231 cells treated with or without NF-$\kappa$B inhibitor PDTC 10 µm for 48 hr (see also *Figure 6—source data 3*, *Figure 6—source data 4*). (**G**) Representative images of the tube formation of HUVEC cells treated with conditional medium from RBM7-depleted MDA-MB-231 in the presence or absence of 10 µm NF-$\kappa$B inhibitor PDTC. Scale bars: 100 µm. Quantification of junctions of endothelial network was conducted by ImageJ software. p values were determined by one-way ANOVA with Tukey's multiple comparison test (n=3). *p<0.05, **p<0.01, ***p<0.001, and ****p<0.0001.

The online version of this article includes the following source data and figure supplement(s) for figure 6:

**Source data 1.** Original file for the western blot analysis in *Figure 6E* (anti-p65, anti-P-p65, anti-I$\kappa$B, anti-P-I$\kappa$B, anti-RBM7, and anti-GAPDH).

**Source data 2.** PDF containing *Figure 6E* and original scans of the relevant western blot analysis (anti-p65, anti-P-p65, anti-I$\kappa$B, anti-P-I$\kappa$B, anti-RBM7, and anti-GAPDH) with highlighted bands and sample labels.

**Source data 3.** Original file for the western blot analysis in *Figure 6F* (anti-p65, anti-P-p65, anti-RBM7, and anti-GAPDH).

**Source data 4.** PDF containing *Figure 6F* and original scans of the relevant western blot analysis (anti-p65, anti-P-p65, anti-RBM7, and anti-GAPDH) with highlighted bands and sample labels.

**Figure supplement 1.** MFGE8-L/S isoforms had no effect on NF-$\kappa$B pathway in breast cancer cells.

**Figure supplement 1—source data 1.** Original file for the RT-PCR analysis of MFGE8 splicing switch upon ASO treatment in *Figure 6—figure supplement 1B*.

**Figure supplement 1—source data 2.** PDF containing *Figure 6—figure supplement 1B* and original scans of the relevant RT-PCR analysis for splicing switch of MFGE8 variants with highlighted bands and sample labels.

**Figure supplement 1—source data 3.** Original file for the western blot analysis in *Figure 6—figure supplement 1E* (anti-p65, anti-P-p65, anti-I$\kappa$B, anti-P-I$\kappa$B, anti-Flag, and anti-GAPDH).

**Figure supplement 1—source data 4.** PDF containing *Figure 6—figure supplement 1E* and original scans of the relevant Western blot analysis (anti-p65, anti-P-p65, anti-I$\kappa$B, anti-P-I$\kappa$B, anti-Flag, and anti-GAPDH) with highlighted bands and sample labels.

**Figure supplement 1—source data 5.** Original file for the western blot analysis in *Figure 6—figure supplement 1F* (anti-p65, anti-P-p65, anti-RBM7, and anti-GAPDH).

**Figure supplement 1—source data 6.** PDF containing *Figure 6—figure supplement 1F* and original scans of the relevant western blot analysis (anti-p65, anti-P-p65, anti-RBM7, and anti-GAPDH) with highlighted bands and sample labels.

and revealed a noticeably positive correlation between the expression of RBM7 and exon7 inclusion of MFGE8 in breast cancer (*Figure 7E*), further supporting MFGE8 isoform switch regulated by RBM7.

## Discussion

Previous reports have recognized the RNA-binding protein RBM7 as a component of the NEXT complex that could facilitate RNA exosome targeting to a subset of non-coding RNAs and aberrant transcripts (*Meola et al., 2016*; *Preker et al., 2008*), including short-lived exosome substrates and pre-mRNAs, for surveillance and turnover. In addition to this specific characteristic, RBM7 was shown to interact with subunits of 7SK snRNP complex and thus activate P-TEF downstream genes in response to DNA damage (*Bugai et al., 2019*). Herein, we discover a novel role of RBM7 in alternative splicing control through directly binding to its cognate cis-element in pre-mRNA. Using whole-transcriptome sequencing, we found the depletion of RBM7 contributed to exon exclusion in approximately 70% SE/A5E events, implying RBM7 mainly functions as a splicing activator in the context of SE/A5E in breast cancer. However, the increase of PSI also occurred due to RBM7 knockdown in a smaller faction of splicing events, such as ZMYND8. We speculated this converse impact may be associated with the intrinsically sophisticated splicing processes driven by distinct core splicing factors, cognate cis-acting element and co-factors in a context-dependent manner (e.g. pre-mRNA structure, base modification, antagonistic or cooperative effect of multiple splicing factors). Similar to many splicing factors, like SR proteins and hnRNPs, RBM7 has an RRM-domain required for target RNA recognition in the N-terminus. We performed co-IP experiment and found that RBM7 could interact with RNA splicing factor SF3B2, a component of spliceosomal U2 SnRNP complex (*Figure 7—figure supplement 1B*). Consistent with the AS regulation of MFGE8 by RBM7, the depletion of SF3B2 also promoted exon7

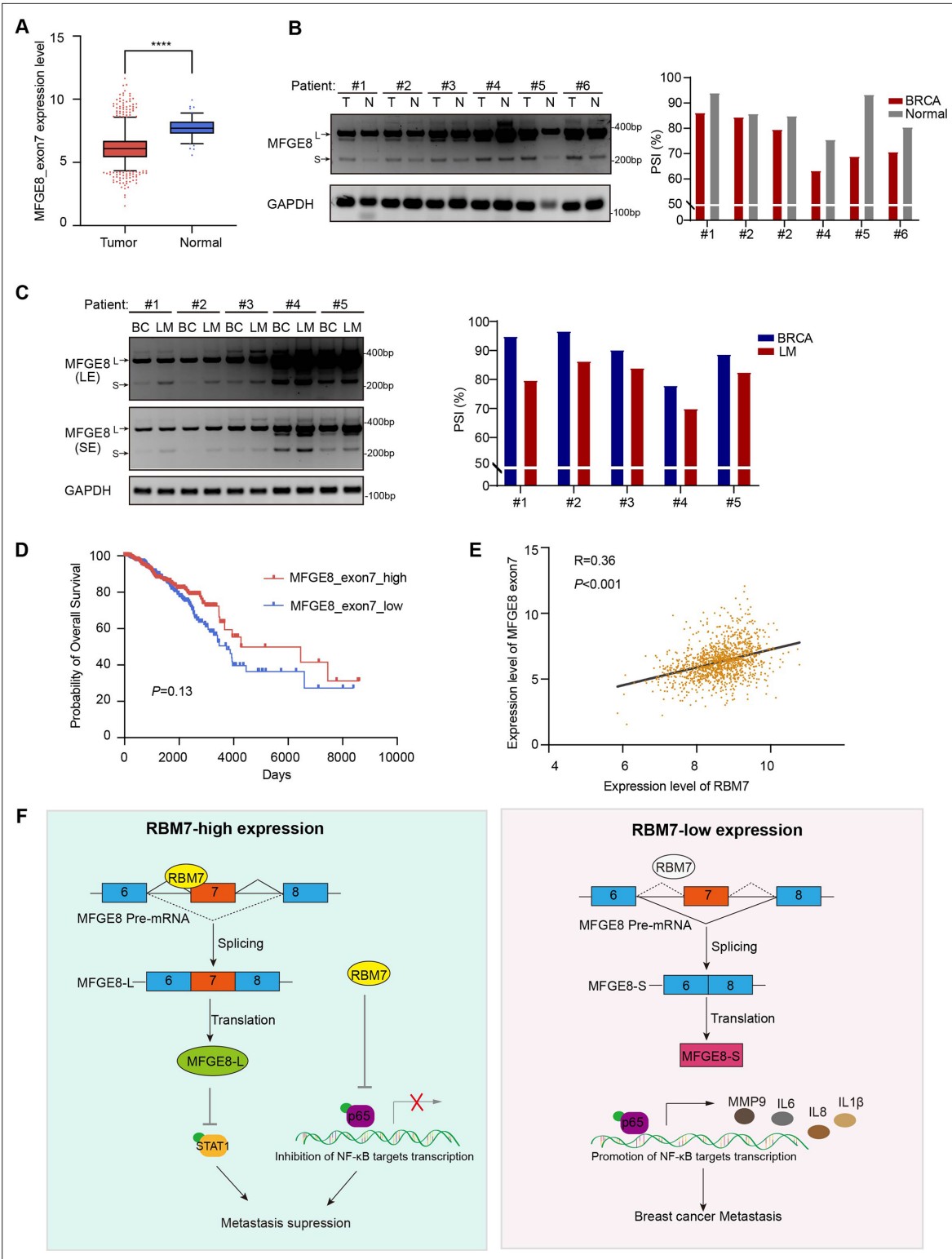

**Figure 7.** Splicing shift of MFGE8 toward exon7 exclusion was negatively correlated with RBM7 expression in patients with breast cancer. (**A**) MFGE8 exon7 expression level was analyzed in breast carcinoma in comparison with normal tissues based on the TCGA dataset. Plotted are the mean ± SD from 114 normal tissues and 1094 tumor tissues, with **** p<0.0001 as determined by unpaired Student' t test. (**B**) The splicing alteration of MFGE8 in 6 pairs of primary breast cancer tissues and adjacent normal tissues was examined using RT-PCR (see also *Figure 7—source data 1*, *Figure 7—source data 2*). The quantification of PSI vales was based on relative band intensities using Image J software. (**C**) The splicing alteration of MFGE8 in primary breast cancer tissues and corresponding lymph node metastases was identified by RT-PCR assays (see also *Figure 7—source data 3*, *Figure 7—source*

*Figure 7 continued on next page*

*Figure 7 continued*

**data 4**). The quantification of PSI vales was determined by Image J software. (**D**) The correlation between the expression of MFGE8 exon7 and overall survival (OS) of breast cancer patients was analyzed based on the TCGA dataset. p value was assessed with Mantel-Cox log-rank test. (**E**) Correlation of RBM7 expression with MFGE8 exon7 levels was analyzed using Pearson's correlation coefficient method. (**F**) The mechanistic model of how RBM7 regulated metastasis of breast cancer through regulating MFGE8 splicing switch.

The online version of this article includes the following source data and figure supplement(s) for figure 7:

**Source data 1.** Original file for the RT-PCR analysis of MFGE8 splicing switch in primary breast cancer tissues and adjacent normal tissues in *Figure 7B*.

**Source data 2.** PDF containing *Figure 7B* and original scans of the relevant RT-PCR analysis for MFGE8 splicing switch with highlighted bands and sample labels in primary breast cancer tissues and adjacent normal tissues.

**Source data 3.** Original file for the RT-PCR analysis of MFGE8 splicing switch in primary breast cancer tissues and corresponding lymph node metastases in *Figure 7C*.

**Source data 4.** PDF containing *Figure 7C* and original scans of the relevant RT-PCR analysis for splicing switch of MFGE8 variants with highlighted bands and sample labels in primary breast cancer tissues and corresponding lymph node metastases.

**Figure supplement 1.** MFGE8 exon7 skipping is enhanced in a variety of cancers.

**Figure supplement 1—source data 1.** Original file for the western blot analysis of Flag-tagged precipitated complexes in *Figure 7—figure supplement 1B* (anti-Flag, anti-HA and anti-GAPDH).

**Figure supplement 1—source data 2.** PDF containing *Figure 7—figure supplement 1B* and original scans of the relevant western blot analysis of Flag-tagged precipitated complexes (anti-Flag, anti-HA, and anti-GAPDH) with highlighted bands and sample labels.

**Figure supplement 1—source data 3.** Original file for RT-PCR analysis of splicing switch of MFGE8 upon SF3B2 knockdown in *Figure 7—figure supplement 1C*.

**Figure supplement 1—source data 4.** PDF containing *Figure 7—figure supplement 1C* and original scans of the relevant RT-PCR analysis of MFGE8 splicing switch upon SF3B2 knockdown with highlighted bands and sample labels.

skipping (*Figure 7—figure supplement 1C-D*), implying a cooperative effect of the two proteins in regulating MFGE8 splicing. This is in concert with a previous study that RRM domain of RBM7 could bind a proline-rich segment within SF3B2 (*Falk et al., 2016*). The interaction mode with strong similarity to RBM7$^{RRM}$–ZCCHC8$^{Proline}$ interaction in the NEXT complex indicated mutually exclusive binding of SF3B2 and ZCCHC8 to RBM7. Thus, RBM7 appears to play dual, but not conflicting, roles during RNA processes depending on its interaction with the spliceosome or exosome.

Currently, the pathobiological function of RBM7 in breast cancer is underexplored except one study that claimed an oncogenic role of RBM7 since its knockdown inhibited breast cancer cell proliferation (*Xi et al., 2020*). Nevertheless, in our study, RBM7 knockdown in breast cancer displayed visibly pro-metastasis function with activated NF-κB signaling. More importantly, according to TCGA database (total cases >1,000 used for analysis), RBM7 shows decreased expression in breast cancer, particularly in the metastasis loci and has positive correlation with patients' survival outcomes. It is therefore highly doubtful that RBM7 could serve as an oncogene in breast cancer as previously reported. Moreover, we can't exclude additional biological function of RBM7 in breast cancer. RBM7-targeted silencing for breast cancer would likely be detrimental for neighbouring heathy tissue and immune environment as a result of NF-κB-dependent pro-inflammatory mediators and cytokines secretion. Other splicing events switch triggered by RBM7 might also be linked to tumor development. For example, we found RBM7 regulated alternative splicing of AKT2, which is involved in glucose metabolism (*Hay, 2011*). Further investigation is warranted to elaborate more splicing mechanisms of RBM7 in breast cancer.

In conclusion, we discovered that down-regulated RBM7 and its cognate elevated exon7-exclusion MFGE8 were tightly related to enhanced breast cancer metastatic potential and poor prognosis of patients. MFGE8-L, the canonical variant, possessed the capability to trigger STAT1 phosphorylation and inhibit the motility and invasion of breast cancer cells, whereas splice switch to MFGE8-S was found loss-of-function for this tumor-suppressive role. At another layer, RBM7 reduction led to NF-κB pathway activation, resulting in enhancement of breast tumor aggressiveness. Overall, our findngs implicated that activated NF-κB pathway worked in concert with MFGE8 splicing transition to spur metastasis in RBM7-depleted breast cancer. Additionally, although the detailed clinical significance underlying their cooperation need further in-depth investigation, we propose that RBM7 and its regulated splicing variants might serve as potential therapeutic biomarker and prognostic indicator in breast cancer patients.

## Materials and methods

### Cell culture

Human breast cancer cell lines (MDA-MB-231, MCF7, HCC1937, and BT-549) and human embryonic kidney cell HEK293T in this article were obtained from the American Type Culture Collection (ATCC). MDA-MB-231 cells were cultured without carbon dioxide at 37°C in Leibovitz's L-15 medium (Gibco) contained 10% FBS (SINSAGE TECH). MCF7 and HEK293T cells were cultured with DMEM mediun (Gibco) supplemented with 10% FBS. HCC1937 cells were cultured with RPMI-1640 media (Gibco) supplemented with 10% FBS, while BT-549 was maintained with RPMI-1640 media (Gibco) supplemented with 10% FBS and 0.023 U/ml insulin. HUVECs were cultured with Endothelial Cell Medium (ScienCell) supplemented with ECGS (ScienCell), P/S solution (ScienCell) and 10% FBS (ScienCell). Except for MDA-MB-231, all the cells above were cultured in a 37 °C humidified incubator.

### Stable breast cancer cell lines construction

To generate RBM7 overexpression vector, its coding regions (NM_016090.4) were cloned into pCDH-CMV-MCS-EF1-Puro with N-terminal Flag tag or HA-tag.To construct MFGE8-L/S vectors, the coding regions of MFGE8-L (NM_005928.4) and MFGE8-S (NM_001114614.3) were cloned into pCDH-CMV-MCS-EF1-Puro with N-terminal Flag tag. For RBM7 knockdown, target-specific shRNA (RBM7-sh1: 5'-CGAAAGGACTATGGATAACAT-3';RBM7-sh2:5'-CTACCTAGCAGATAGACATTA-3') was cloned into pLKO.1 vector. And HEK293T cells were transfected these over-expression (or low-expression) lentivirus plasmisd together with lentivirus packaging plasmids pPAX2 and pMD2 according to the manufacturer's instructions. Viruses were collected 48 hr after transfection and then to infect breast cancer cell line with 8 µg/ml polybrene (beyotime biotechnology) for 24 hr. Positive cells were selected with 3 µg/ml puromycin.

### Western blot assay

Cells were lysed with RIPA buffer (10 mM Tris-HCl pH 7.5, 150 mM NaCl, 0.1% SDS, 1% Triton X-100, and 1% sodium deoxycholate) supplemented with protein tyrosine phosphatase inhibitor $Na_3VO_4$ (1 mM), serine protease inhibitor PMSF (1 mM) and protease inhibitor Cocktail. Immunoblotting assay was performed using standard methods. The following antibodies were used: anti-RBM7 (Sigma-Aldrich, HPA013993), anti-Flag (Sigma-Aldrich, F1804), anti-GAPDH (Proteintech, 10494–1-AP), anti-α-Tubulin (Proteintech, 11224–1-AP), anti-HA tag (Abcam, ab9110), anti-Phospho-NF-κB p65 (Cell Signaling Technology, #3033), anti-NF-κB p65 (Cell Signaling Technology, #8242), anti-IκBα (Cell Signaling Technology, #4814), anti-Phospho-IκBα (Ser32) (Cell Signaling Technology, #2859), anti-Phospho-stat1 (Tyr701) (Cell Signaling Technology, #8062), and anti-MFGE8 (R&D systems, MAB2767). The protein bands were detected with ECL enhanced chemiluminescence regent kits (NCM, Biotech), visualized using the MiniChemi Cheminescent image system (Beijing Sage Creation, China).

### RNA isolation, RNA sequencing, and PCR

Total RNA from cells and tumor tissues was extracted using Trizol (Invitrogen) according to the manufacturer's instructions. Total mRNA was cleaned up using RNeasy Lipid Tissue Mini Kit (QIAGEN, USA). And RNA-seq was carried out on platform of Novogene Illumina sequencing. cDNAs were obtained using *Evo M-MLV* Plus 1st Strand cDNA Synthesis Kit (Accurate Biotechnology, AG11615) with random primers. qRT-PCR was carried out using MonAmp Universal ChemoHS Specificity Plus qPCR Mix. GAPDH mRNA was examined as control for normalization. Gene expression relative to GAPDH was calculated using the formula $2^{-\Delta\Delta CT}$. For identified alternative splicing events, primers of the upstream and downstream of variable exons or introns were designed and PCR assay was performed. PCR products were subject to DNA agarose gel electrophoresis. AS events change of percent-spliced-in (PSI) values were the ratio of retained splices to total transcripts. The amount of each splicing isoform was measured with ImageJ. All the primers are showed in *Supplementary file 1*.

### Migration and invasion assays

For transwell migration assay, $1 \times 10^5$ MDA-MB-231 or other breast cancer cells (MCF7, BT-549 and HCC1937) were plated in the upper chamber of 6.5 mm insert (Costar, USA) with 8.0 µm polycarbonate membrane with 100 µL serum-free medium. For invasion assay, $1 \times 10^5$ breast cancer cells

were seeded in the top chamber of each Matrigel-coated insert (BD Biosciences, USA). After 24 hr or 48 hr of incubation at 37 °C, migrated or invaded cells were fixed with 4% PFA and stained in 0.1% crystal violet. The cell number was counted by ImageJ software in three views from three independent experiments.

## Wound healing

Knockdown RBM7 in MDA-MB-231 and 4T1 cells or control cells were implanted into 6-well culture dishes. When the cells grew about 90% confluence, lines of the same width were drawn on the bottom of the culture dish using autoclaved sterile tips. Images were captured at 0 and 36 hr after the wounding. Data shown were representative of three independent repeats.

## Gelatin degradation assay

To evaluate the ability of breast cancer cells to form invadopodia and degrade matrix, QCM Gelatin Invadopodia Assay (ECM670-1, Merck Millipore) was performed according to the manufacturers' protocol. Briefly, $1 \times 10^4$ MFGE8-L/S-overexpressing or control MDA-MB-231 cells were suspended in L-15 medium with 10% FBS and seeded into the pre-treated 96-well plate. After 48 hr, the cells were washed twice with PBS. Fix cells with 3.7% formaldehyde for 30 min. Then these cells were stained respectively with TRITC-phalloidin and DAPI for 1 hr. The ability of cells to degrade the matrix was imaged using immunofluorescence microscopy (LEICA DMi8, Germany) and the area of invadopodia was measured using the Image J software.

## Tube formation of HUVECs assay

Capillary tube formation assay was used to observe angiogenesis of HUVECs. HUVECs were seeded into 48-well plate coated with growth factor reduced Matrigel (BD Biosciences). HUVECs cells were treated with conditioned medium for 4–5 hr and then formation of capillary-like tubes was observed and stained with calcein for 30 min. Images of blood vessels were taken using fluorescence microscopy. And the tube length was analyzed by ImageJ software.

## Immunofluorescence assay

After seeded on coverslips for 36 hr, the MDA-MB-231 cells were immersion fixed with 4% PFA for 20 min at room temperature, washed three times in PBS. Then the cells were treated with 0.2% Triton-X100 in PBS for 10 min and blocked with 3% BSA for 10 min. Next the cells were immersed in primary antibody at 4°C overnight, washed in PBS three times for 5 min each, and incubated with secondary antibody labeled with Alexa fluorophore 488/592 for 1 hr at RT. After washing three times in PBS, the cells were dyed with DAPI. In addition, antibodies targeting RBM7 (Sigma, HPA013993), and Flag (Sigma, 3165) were used in the assay. The pictures are captured with fluorescent microscope (Leica, USA).

## RNA immunoprecipitation assay

RNA immunoprecipitation was performed as previously described. In brief, HEK293T cells were transfected Flag-RBM7 or control plasmid. After 48 hr of transfection, Flag-RBM7 or control cells were collected and cross-linked by 1% formaldehyde, homogenized in lysis buffer (50 mM Tris-HCl Ph 7.5, 0.4 M NaCl, 1 mM EDTA, 1 mM DTT, 0.5% TritonX-100, 10% Glycerol supplemented with RNase inhibitor and broad-spectrum protease inhibitors Cocktail), and broke with an ultrasonic crusher. And solubilized cell lysate was obtained and precleared by ProteinG-agarose beads together with nonspecific tRNA to remove nonspecific binding. Then binding RNAs with RBM7 protein were precipitated with Anti-Flag M2 Affinity beads. Co-precipitated RNAs were subject to RT-PCR. Input controls (total RNAs) and Flag-controls were assayed simultaneously to prove that the detected signals were the result of RNAs specifically binding to RBM7. The gene-specific primers were designed MFGE8-RIP1-F:5'-tatttatgccccctcaccgc-3'; MFGE8-RIP1-R:5'-atggtggctgcctgtaat-3';MFGE8-RIP2-F:5'-ctgaggctcacccaagtag-3';MFGE8-RIP2-R:5'-GTAGGATGCCACAAACTG-3'. GAPDH mRNA was as negative control (GAPDH-F 5'-GGGGAAGGTGAAGGTCG-G-3'; GAPDH-R 5'-TTGAGGTCAATG AAGGGG-3').

### Antisense oligonucleotide (ASO)-mediated splicing blockade

MFGE8-ASOs were synthesized with a full-length phosphorothioate backbone and 2'-O-methyl modified ribose molecules (GenScriptBiotech Corp). ASO was transfected at final concentration of 500 nM using Lipofectamine3000 (Thermo Fisher Scientific) according to the manufacturer's instructions. The MCF7 cells were harvested for RNA and splicing assays 48 hr after transfection.

### Animal experiments

All procedures and experimental protocols involving mice were approved by the Institutional Animal Care and Use Committee (IACUC) of Dalian Medical University (approval no. AEE21015). Female BALB/C mice (6 weeks old, 4–5 mice per group as indicated in figures and/or figure legends) were injected into tail vein with murine 4T1 breast cancer cells ($6\times10^5$ cells) to develop mice model of pulmonary metastasis. After about 3 weeks, all mice were euthanized, and lungs were harvested, fixed and paraffin embedded. After lungs were H&E stained, the number of metastatic lesions were enumerated by pathologist using brightfield microscopy.

### Clinical tissues samples collection

All human breast cancer tissues or normal tissue were obtained with written informed consent from patients or their guardians prior to participation in the study. The Institutional Review Board of the Dalian Medical University approved use of the tumor specimens in this study (approval no.PJ-KS-KY-2022–208).

### Bioinformatics analysis

RBM7 expression data and survival data (including OS and DFS) were obtained from the METABRIC database using the Cbioportal web tool. Based on the expression level of RBM7, the patients were divided into four categories (i.e. Q1, Q2, Q3, Q4) by quartiles, and then log-rank was employed to compare the survival differences of patients in different categories. Cbioportal cited (Cerami E et al. The cBio cancer genomics portal: an open platform for exploring multidimensional cancer genomics data. Cancer Discov. 2012 May;2 (5):401–4). Through the analysis of alternative splicing events in RBM7 knocked-down MDA-MB-231 cells compared to control, exon skipping events were screened out, and then selected the sequence of the 500nt region upstream of the alternative exon for subsequent base sequence enrichment analysis. The XSTREME tool provided by the online tool meme-suite (https://meme-suite.org/meme/) was used to perform Motif Discovery and Enrichment Analysis. E-value <0.05 as the significance standard to output the results.

### Statistical analyses

Statistical analyses and statistical graphs were conducted by GraphPad Prism 8 software. To compare the differences between the two variables, Student's T-test, one-way ANOVA with Dunnett multiple comparisons, and one-way ANOVA with Tukey's multiple comparisons according to the specific situation were applied. $^*p < 0.05$, $^{**}p < 0.01$, $^{***}p < 0.001$, and $^{****}p < 0.0001$ indicated the significant difference.

### Study Approval

The Institutional Animal Care and Use Committee of the Dalian Medical University approved use of animal models in this study (approval no. AEE21015). All human tumor tissues were obtained with written informed consent from patients or their guardians prior to participation in the study. The Institutional Review Board of of the First Affiliated Hospital of Dalian Medical University approved use the tumor specimens in this study (approval no. PJ-KS-KY-2022-208).

## Acknowledgements

This work was supported by the National Key R&D Program of China (2022YFA1104002, 2023YFE0117500 to YW); the National Natural Science Foundation of China (82225034, 81830088 to

YW; 82103148 to YQ; 81872247 to WZ); the Science and Technology Innovation Foundation of Dalian (2022JJ11CG009 to YW), Liaoning Revitalization Talents Program (XLYC2202027 to YW); Interdisciplinary Research Cooperation Project Team Funding of Dalian Medical University (JCHZ2023008 to YQ); Joint Research Program of Natural Science Foundation of Liaoning, China (2023-MSLH-023 to YQ); the Science and Technology Innovation Talent Support Program of Dalian (2022RQ056 YQ); Dalian High Level Talents Renovation Supporting Program (2019RQ097 to WZ).

## Additional information

### Funding

| Funder | Grant reference number | Author |
| --- | --- | --- |
| National Key Research and Development Program of China | 2022YFA1104002 | Yang Wang |
| National Natural Science Foundation of China | 82103148 | Yangfan Qi |
| National Natural Science Foundation of China | 82225034 | Yang Wang |
| Natural Science Foundation of Liaoning Province | 2023-MSLH-023 | Yangfan Qi |
| Dalian Science and Technology Innovation Fund | 2022RQ056 | Yangfan Qi |
| National Natural Science Foundation of China | 81830088 | Yang Wang |
| National Key Research and Development Program of China | 2023YFE0117500 | Yang Wang |
| National Natural Science Foundation of China | 81872247 | Wenjing Zhang |
| Dalian Science and Technology Innovation Fund | 2022JJ11CG009 | Yang Wang |
| Dalian High-Level Talent Innovation Program | 2019RQ097 | Wenjing Zhang |
| Liaoning Revitalization Talents Program | XLYC2202027 | Yang Wang |
| Interdisciplinary Research Cooperation Project Team Funding of Dalian Medical University | JCHZ2023008 | Yangfan Qi |

The funders had no role in study design, data collection and interpretation, or the decision to submit the work for publication.

### Author contributions

Fang Huang, Data curation, Project administration; Zhenwei Dai, Jinmiao Yu, Kainan Wang, Dan Chen, Project administration; Chaoqun Chen, Software, Visualization; Jinrui Zhang, Formal analysis; Jinyao Zhao, Methodology; Mei Li, Software; Wenjing Zhang, Funding acquisition, Methodology; Xiaojie Li, Writing – review and editing; Yangfan Qi, Yang Wang, Conceptualization, Writing – original draft, Writing – review and editing

### Author ORCIDs

Yangfan Qi ⬡ http://orcid.org/0009-0004-9185-7225

Yang Wang http://orcid.org/0000-0001-9385-7393

## Ethics

The Institutional Review Board of the First Affiliated Hospital of Dalian Medical University approved use the tumor specimens in this study (approval no. PJ-KS-KY-2022-208).

This study was performed in strict accordance with the recommendations in the Guide for the Care and Use of Laboratory Animals of the National Institutes of Health. The Institutional Animal Care and Use Committee of the Dalian Medical University approved use of animal models in this study (approval no. AEE21015). All of the animals were handled according to approved institutional animal care and use committee (IACUC) protocols.

Reviewer #1 (Public Review): https://doi.org/10.7554/eLife.95318.3.sa1
Reviewer #2 (Public Review): https://doi.org/10.7554/eLife.95318.3.sa2
Author response https://doi.org/10.7554/eLife.95318.3.sa3

## Additional files

### Supplementary files
• Supplementary file 1. list of primers.
• MDAR checklist

### Data availability

Sequencing data have been deposited in GEO under accession codes GSE248938.

The following dataset was generated:

| Author(s) | Year | Dataset title | Dataset URL | Database and Identifier |
|---|---|---|---|---|
| Huang F, Dai Z, Yu J, Wang K, Chen C, Chen D, Zhang J, Zhao J, Li M, Zhang W, Li X, Qi Y, Wang Y | 2024 | RBM7 regulates breast cancer progression | https://www.ncbi.nlm.nih.gov/geo/query/acc.cgi?acc=GSE248938 | NCBI Gene Expression Omnibus, GSE248938 |

The following previously published dataset was used:

| Author(s) | Year | Dataset title | Dataset URL | Database and Identifier |
|---|---|---|---|---|
| Fukushima K, Motooka D, Nakamura S, Sugihara F, Satoh T | 2020 | Next Generation Sequencing Facilitates Quantitative Analysis of Wild Type and RBM7-/- HEK293 cell lines Transcriptomes and RBM7 targetted RNAs | https://www.ncbi.nlm.nih.gov/geo/query/acc.cgi?acc=GSE144075 | NCBI Gene Expression Omnibus, GSE144075 |

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
