## [Editor Report · eLife assessment]

This study presents a rather **valuable** finding on the RBM7 function in spicing regulation and uncharacterized role of MFGE8 splicing alteration in breast cancer metastasis. The evidence supporting the claims of the authors is **solid**. The work will be of broad interest to clinicians, medical researchers and scientists working on breast cancer.

---

## [Referee Report · Reviewer #1 (Public Review)]

Fang Huang et al found that RBM7 deficiency promotes metastasis by coordinating MFGE8 splicing switch and NF-kB pathway in breast cancer by utilizing clinical samples as well as cell and tail vein injection models.

This study uncovers a previously uncharacterized role of MFGE8 splicing alteration in breast cancer metastasis, and provides evidence supporting RBM7 function in splicing regulation. These findings facilitate the mechanistic understanding of how splicing dysregulation contributes to metastasis in cancer, a direction that has increasingly drawn attention recently, and provides a potentially new prognostic and therapeutic target for breast cancer.

---

## [Referee Report · Reviewer #2 (Public Review)]

In this manuscript, the authors reported the biological role of RBM7 deficiency in promoting metastasis of breast cancer. They further used a combination of genomic and molecular biology approaches to discover a novel role of RBM7 in controlling alternative splicing of many genes in cell migration and invasion, which is responsible for the RBM7 activity in suppressing metastasis. They conducted an in-depth mechanistic study on one of the main targets of RBM7, MFGE8, and established a regulatory pathway between RBM7, MFGE8-L/MFGE8-S splicing switch, and NF-κB signaling cascade. This link between RBM7 and cancer pathology was further supported by analysis of clinical data.

Overall, this is a very comprehensive study with lots of data, and the evidence is consistent and convincing. Their main conclusion was supported by many lines of evidence, and the results in animal models are pretty impressive.

---

## [Author Response]

The following is the authors’ response to the original reviews.

**Reviewers’ Comments:**

**Reviewer #1 (Remarks to the Author):**
Summary:Fang Huang et al found that RBM7 deficiency promotes metastasis by coordinating MFGE8 splicing switch and NF-kB pathway in breast cancer by utilizing clinical samples as well as cell and tail vein injection models.Strengths:This study uncovers a previously uncharacterized role of MFGE8 splicing alteration in breast cancer metastasis, and provides evidence supporting RBM7 function in splicing regulation. These findings facilitate the mechanistic understanding of how splicing dysregulation contributes to metastasis in cancer, a direction that has increasingly drawn attention recently, and provides a potentially new prognostic and therapeutic target for breast cancer.

We thank the reviewer for appreciating the novelty and importance of this study, and have provided new data to address the following concerns raised by the reviewer.

Weaknesses:This study can be strengthened in several aspects by additional experiments or at least by further discussions. First, how RBM7 regulates NF-kB, and how it coordinates splicing and canonical function as a component of NEXT complex should be clarified. Second, although the roles of MFGE8 splicing isoforms in cell migration and invasion have been demonstrated in transwell and wound healing assays, it would be more convincing to explore their roles in vivo such as the tail vein injection model. Third, the clinical significance would be considerably improved, if the therapeutic value of targeting MFGE8 splicing could be demonstrated.

We’re thankful for the constructive suggestions. A preliminary study on the mechanism by which RBM7 regulates NF-kB pathway is already underway. We found RBM7 depletion remarkably promoted the expression of IL-1β as judged by qPCR and ELISA assays (new Figure S5G- S5I, also see below). IL-1β, commonly known as a pro-inflammatory cytokine, could bind to IL-1R and initiate a multistage enzymatic reaction that triggers the activation of NF-κB pathway (Axel Weber, 2010) (Qing Guo, 2024). Thus we speculated that the upregulation of IL-1β might be a causal factor in RBM7-depletion-induced activation of NF-kB signaling. It will be interesting to determine the complete molecular mechanism in our future study. In addition, we performed a co-IP experiment and found that RBM7 could interact with RNA splicing factor SF3B2, a component of spliceosomal U2 snRNP complex (new Figure S6B, also see below). Consistent with the AS regulation of MFGE8 by RBM7, the depletion of SF3B2 also promoted exon7 skipping, implying a cooperative effect of the two proteins in regulating MFGE8 splicing (new Figure S6C-6D, also see below). This is in concert with a previous study that RRM domain of RBM7 could bind a proline-rich segment within SF3B2 (Falk, Finogenova et al., 2016). The interaction mode with strong similarity to RBM7RRM–ZCCHC8Proline interaction in the NEXT complex indicated mutually exclusive binding of SF3B2 and ZCCHC8 to RBM7. Thus, RBM7 appears to play dual, but not conflicting, roles during RNA processes depending on its interaction with the spliceosome or exosome (see line 427-437 in the new manuscript).

**Author response image 1. sa3fig1:** The mRNA levels of IL-1β in MDA-MB-231 or BT549 cells with stable RBM7 knockdown or control vector were examined by qRT-PCR approach.

**Author response image 2. sa3fig2:** Supernatants from RBM7-knockdown MDA-MB-231 or BT549 cells were collected and protein expression of IL-1β was measured by ELISA kit.

**Author response image 3. sa3fig3:** The knockdown efficiency of RBM7 in two breast cancer cell lines were determined by qRT-PCR approach.

**Author response image 4. sa3fig4:** Immunoprecipitation assay was performed in breast cancer cells expressing HA-RBM7 and Flag-SF3B2 or empty vector. The Flag-tagged precipitated complexes and lysates were analyzed through western blotting.

**Author response image 5. sa3fig5:** The splicing shift of MFGE8 upon SF3B2 knockdown in breast cancer cells was examined by RT-PCR approach. The mean ± SD of PSI values derived from three independent replicates is shown.

**Author response image 6. sa3fig6:** The SF3B2 knockdown efficiency was examined by qRT-PCR.

To further corroborate the roles of two MFGE8 isoforms in cell invasion, we have performed Fluorescent Gelatin Degradation Assays for investigating invadopodia formation. Consistent with the transwell assay results, MFGE8-L up-regulation suppressed breast cancer cells invasion through a layer of extracellular matrix, whereas breast cancer cells with ectopic expression of MFGE8-S acquired enhanced ability to degrade matrix and invasion (new Figure 5B, also see below). In addition, to determine the therapeutic value of targeting MFGE8 splicing, we transfected triple-negative breast cancer cells with ASOs targeting RBM7-binding motif and examined the potential impact on cell aggressiveness. The results showed an obvious increase in exon7-skipped variant of MFGE8 as compared to the scramble negative control ASOs, meanwhile, the migrative and invasive ability of breast cancer cells treated with splice-targeting ASOs was significantly boosted (new Figure 6B and S5B, also see below), further suggesting that RBM7-knockdown stimulated aggressiveness of breast cancer at least partially relies on splicing switch of MFGE8.

**Author response image 7. sa3fig7:** Gelatin degradation assay was performed to test the effect of RBM7 knockdown on invadopodia function. 10000 cells were plated onto FITC-gelatin substrates (Green) and cultured for 48 h. Representative images are shown (red, Cy3-phalloidin; blue, DAPI) and the degraded areas were quantified by Image J software. Scar bars = 50 μm. P values were determined by one-way ANOVA with Tukey's multiple comparison test (n = 3).

**Author response image 8. sa3fig8:** Representative transwell analysis of migrative/invasive capability of breast cancer cells transfected with 500 nM ASO directed against RBM7-binding region in MFGE8 pre-mRNA. P values were determined by one-way ANOVA with Tukey's multiple comparison test.

**Author response image 9. sa3fig9:** RT-PCR quantification of two MFGE8 isoforms after transfecting breast cancer cells with 500 nM ASO directed against RBM7-binding region in MFGE8 pre-mRNA. P values were calculated by one-way ANOVA with Tukey's multiple comparison test.

The minor concerns(1) Several figure legends do not match with the images, for example, Figure 2K, Figure 4, Figure 7D, and 7E, and the description of Fiure 7F is missing in the text.

As suggested by the reviewer, we have checked all of the figure legends carefully and corrected all of the misinterpretation.

(2) The statistical methods for Figure1A and Figure1B should be indicated.

As suggested by the reviewer, we have included the statistical methods for Figure1A and 1B in Figure1 legend. Data in Figure 1A and 1B are presented as means ± SD and *P* values were obtained by Mantel-Cox log-rank test.

(3) The molecular weight of the proteins in the Western Blot images should be marked.

As suggested by the reviewer, we have added the molecular weight of proteins in all of the western blot images.

(4) The sequences where RBM7 binds on MFGE8 RNA should be clearly indicated.

We thank the reviewer for this question. We analyzed the sequence of alternative exon 7 and the motifs nearby its 5’ or 3’ splice sites, and found two RBM7 potentially binding motifs are positioned in proximal to the pseudo 3’ splice site. Subsequent RT-PCR for the precipitation in RIP assays confirmed RBM7 could bind to the upstream sequence containing 5’-UUUCUU-3’ motifs adjacent to intron6/exon7 junction of MFGE8 cassette exon, but not another region nearby it. To pinpoint the location for the potential cis-element for AS regulation by RBM7, we designed antisense oligonucleotides (ASOs) to block RBM7 potentially binding sites (UUUCUU). As shown in revised Figure 4F, when compared to scramble ASO, targeting ASOs contributed to the exclusion of exon7. Additionally, we constructed an exogenous MFGE8 splicing reporter containing exon 6-8 and partial intron sequences to determine the binding site for AS regulation by RBM7. The depletion of RBM7 still induced the splicing shift of the minigene reporter by elevating MFGE8-S variant. While the binding motif UUUCUU was removed or mutated, RBM7 failed to affect the splicing outcomes of MFGE8 (new Figure S3C, also see below). Due to its close proximity to 3’ splice site, UUUCUU residues bound by RBM7 is very likely to participate in spliceosome assembly at the upstream 3’ splice site of exon7, which may explain why disruption of the motif led to almost complete exon7 skipping. The above data suggested that RBM7 regulated the exon skipping of MFGE8 by binding to UUUCUU located six nucleotides upstream of the 3’ splice-site of exon7.

**Author response image 10. sa3fig10:** Upper: the red line in diagram indicates ASOs targeting region which contains UUUCUU; down: MCF7 and MDA-MB-231 cells were transfected with ASOs targeting MFGE8 pre-mRNA for 48h and then applied for RT-PCR identification. P values were determined by one-way ANOVA with Tukey's multiple comparison test.

**Author response image 11. sa3fig11:** Upper: MFGE8 min-splicing reporters with mutation in the RBM7 binding site or a non-specific binding were generated and shown in cartoon; down: RT-PCR assays were performed to identify the splicing outcomes of MFGE8 reporter while RBM7 was depleted in breast cancer cells.

(5) Some typos, graphic errors, and sentences are hard to understand and need to be corrected, such as lines 80-81, 249-250, line 221 "motfs", line 319 "RBM4". Please carefully proofread and revise the entire manuscript.

As suggested by the reviewer, we have corrected typos and graphic errors mentioned above. In addition, this manuscript was also extensively edited to improve grammar and sentence structure.

(6) Define the abbreviations when they first appear, such as MFGE8-L, RBM, etc.

We thank the reviewer for raising this point. We have defined the abbreviations when firstly presented in the manuscript.

**Reviewer #2 (Public Review):**
Summary:In this manuscript, the authors reported the biological role of RBM7 deficiency in promoting metastasis of breast cancer. They further used a combination of genomic and molecular biology approaches to discover a novel role of RBM7 in controlling alternative splicing of many genes in cell migration and invasion, which is responsible for the RBM7 activity in suppressing metastasis. They conducted an in-depth mechanistic study on one of the main targets of RBM7, MFGE8, and established a regulatory pathway between RBM7, MFGE8-L/MFGE8-S splicing switch, and NF-κB signaling cascade. This link between RBM7 and cancer pathology was further supported by analysis of clinical data.Strengths:Overall, this is a very comprehensive study with lots of data, and the evidence is consistent and convincing. Their main conclusion was supported by many lines of evidence, and the results in animal models are pretty impressive.Weaknesses:However, there are some controls missing, and the data presentation needs to be improved. The writing of the manuscript needs some grammatical improvements because some of the wording might be confusing.

We thank the reviewer for the positive comments on this work, and have addressed all the concerns raised by the reviewer.

Specific comments:(1) Figure 2. The figure legend is missing for Figure 2C, which caused many mislabels in the rest of the panels. The labels in the main text are correct, but the authors should check the figure legend more carefully. Also in Figure 2C, it is not clear why the authors choose to examine the expression of this subset of genes. The authors only refer to them as "a series of metastasis-related genes", but it is not clear what criteria they used to select these genes for expression analysis.

We thank the reviewer for raising this question. We have included the figure legend for Figure 2C and improved other figure legends throughout the article. For the second question, since gene ontology analysis of RNA-seq data in RBM7-depleted breast cancer cells showed that a series of differentially expressed genes were enriched in metastasis-associated processe, we identified the expression of this subset of genes in breast cancer cells in the presence or absence of RBM7 by heatmap differential analysis based on qRT-PCR results. To clarify this point and address the reviewer’s concern, we have improved the relevant description of this part (see line 174-180 in the new manuscript).

(2) Line 218-220. The comparison of PSI changes in different types of AS events is misleading. Because these AS events are regulated in different mechanisms, they cannot draw the conclusion that "the presence of RBM7 may promote the usage of alternative splice sites". For example, the regulators of SE and IR may even be opposite, and thus they should discuss this in different contexts. If they want to conclude this point, they should specifically discuss the SE and A5SS rather than draw an overall conclusion.

We are thankful for the reviewer’s valuable comment. According to the suggestion, we have removed the overall conclusion and corrected to discuss in SE and A5SS.

(3) In the section starting at line 243, they first referred to the gene and isoforms as "EFG-E8" or "EFG-E8-L", but later used "EFGE8" and "EFGE8-L". Please be consistent here. In addition, it will be more informative if the authors add a diagram of the difference between two EFGE8 isoforms in terms of protein structure or domain configuration.

As suggested by the reviewer, we keep using the name “MFGE8-L” for the canonical MFGE8 isoform and “MFGE8-S” for the truncated isoform in this manuscript. In addition, to clarify the structural basis for the different tumor invasion-related functions of two MFGE8 isoforms, we have included a diagram of their domain configuration in new Figure S4F and predicted protein structure in new Figure S4G. The details in the revised manuscript are given below:

**Author response image 12. sa3fig12:** Schematic diagram of the domain composition of two MFGE8 isoforms. Upper: the full-length variant with exon7 indicated by yellow square; down: the truncated variant with exon7 skipping.

**Author response image 13. sa3fig13:** The model structure of two MFGE8 isoforms was implemented using SwissModel software. The F5/8 type C2 protein domain excluded from MFGE8-S variant was marked in red.

(4) Figure 7B and 7C. The figures need quantification of the inclusion of MFGE exon7 (PSI value) in addition to the RT-PCR gel. The difference seems to be small for some patients.

As suggested by the reviewer, we have included the relative quantification of PSI for endogenous MFGE8 in breast cancer patients and found increased proportion of exon7 exclusion in most tumor samples when compared to normal tissues (case#1: 86:94; case#2: 84:86; case#3: 79:85; case#4: 63:75; case#5: 69:93; case#6: 71:80) (new Figure 7B, also see below). On the other hand, we have expanded the number of metastatic breast cancer cases and quantified the the AS events within MFGE8 by analyzing the PSI values. The lymph node metastases contain a higher proportion of MFGE8 variant with skipped exon7 in comparison with paired primary tumor tissues (case#1: 80:95; case#2: 86:97; case#3: 84:90; case#4: 70:78; case#5: 83:89) (Figure 7C). This is coherent with decreased RBM7 expression levels found in breast cancer with lymph node metastasis.

**Author response image 14. sa3fig14:** The splicing alteration of MFGE8 in 6 pairs of primary breast cancer tissues and adjacent normal tissues was examined using RT-PCR. The quantification of PSI vales was based on relative band intensities using Image J software.

**Author response image 15. sa3fig15:** The splicing alteration of MFGE8 in primary breast cancer tissues and corresponding lymph node metastases was identified by RT-PCR assays. The quantification of PSI vales wa determined by Image J software.

Minor comments:The writing in many places is a little odd or somewhat confusing, I am listing some examples, but the authors need to polish the whole manuscript more to improve the writing. 1. Line 169-170, "...followed by profiling high-throughput transcriptome by RNA sequencing", should be "followed by high-throughput transcriptome profiling with RNA sequencing". 2. Line 170, "displayed a wide of RBM7-regulated genes were enriched...", they should add a "that" after the "displayed" as the sentence is very long. 3. Line 213, "PSI (percent splicing inclusion)" is not correct, PSI stands for "percent spliced in". 4. Line 216-217, the sentence is long and fragmented, they should break it into two sentences. 5. Line 224, the "tethering" should be changed to "recognizing". There is a subtle difference in the mechanistic implication between these two words. 6. Line 250, should be changed to "...in the ratio of two MFGE8 isoforms".

We thank the detailed comments from the reviewer. The points mentioned above has been addressed one by one and this manuscript was also extensively edited to improve grammar and sentence structure for better understanding.

**References**

Axel Weber PW, Michael Kracht* (2010) Interleukin-1 (IL-1) Pathway. SCIENCESIGNALING.

Qing Guo1, Yizi Jin1,2, Xinyu Chen3, Xiaomin Ye4, Xin Shen5, Mingxi Lin1,2, Cheng Zeng1,2, Teng Zhou1,2 and Jian Zhang1,2 (2024) NF-κB in biology and targeted therapy: new insights and translational implications. Signal Transduction and Targeted Therapy.

Falk S, Finogenova K, Melko M, Benda C, Lykke-Andersen S, Jensen TH, Conti E (2016) Structure of the RBM7–ZCCHC8 core of the NEXT complex reveals connections to splicing factors. Nature Communications.